# PatCID: an open-access dataset of chemical structures in patent documents

Lucas Morin [1,2] ✉, Valéry Weber[1], Gerhard Ingmar Meijer[1], Fisher Yu[2] & Peter W. J. Staar[1] ✉

The automatic analysis of patent publications has potential to accelerate research across various domains, including drug discovery and material science. Within patent documents, crucial information often resides in visual depictions of molecule structures. PatCID (Patent-extracted Chemical-structure Images database for Discovery) allows to access such information at scale. It enables users to search which molecules are displayed in which documents. PatCID contains 81M chemical-structure images and 14M unique chemical structures. Here, we compare PatCID with state-of-the-art chemical patent-databases. On a random set, PatCID retrieves 56.0% of molecules, which is higher than automatically-created databases, Google Patents (41.5%) and SureChEMBL (23.5%), as well as manually-created databases, Reaxys (53.5%) and SciFinder (49.5%). Leveraging state-of-the-art methods of document understanding, PatCID high-quality data outperforms currently available automatically-generated patent-databases. PatCID even competes with proprietary manually-created patent-databases. This enables promising applications for automatic literature review and learning-based molecular generation methods. The dataset is freely accessible for download.

Recent advances in document understanding enable the acceleration of discoveries in chemistry. Patent documents and scientific publications provide a wealth of knowledge that can only be effectively exploited by automated large-scale processing. Searching information from patent documents is at high stakes for industrial applications, especially with respect to freedom-to-operate, prior-art search, or landscape analysis[1]. Additionally, in the chemistry domain, a substantial proportion of scientific findings is disseminated only in patent documents, or only later published in scientific journals[2–4]. With the continuous growth of patent applications per year, access to chemical information in patent documents poses key challenges. Chemical knowledge, including compounds, reactions, and molecular properties, are presented in documents in a non-standardized way, using multiple modalities such as text descriptions, tables, and depictions. Proprietary databases, such as Elsevier Reaxys[5] and CAS SciFinder[n6], aim to provide a solution to search this chemical information in documents. Being manually curated, they are considered the gold-

standard for literature search. However, their development requires massive and continuous effort, and given the manual process, they cannot cover all patent documents and collections. To address these challenges, several projects leveraging automatic document processing have been developed, including SureChEMBL[7], Google Patents SciWalker, Patentscope[8], or IBM SIIP[9]. In recent years, new pipelines have also been developed to convert documents into chemical structures[10]. The currently available evaluations suggest that these databases fall short in comparison to manually-processed databases, both in terms of document coverage and processing quality[1,11,12]. Especially, documents published before 2000 or from the Asian Pacific patent offices are not covered, or with poor quality, while they provide unique and disruptive innovation[13]. Furthermore, manually- and automatically-created databases are designed to retrieve a set of document identifiers referring to a specified molecule. Users then need to open the documents and manually search for pages containing references to the molecule they are interested in. This approach does

[1]IBM Research, Säumerstrasse 4, 8803 Rüschlikon, Switzerland. [2]Department of Information Technology and Electrical Engineering, ETH Zürich, Sternwartstrasse 7, 8092 Zürich, Switzerland. ✉e-mail: lum@zurich.ibm.com; taa@zurich.ibm.com

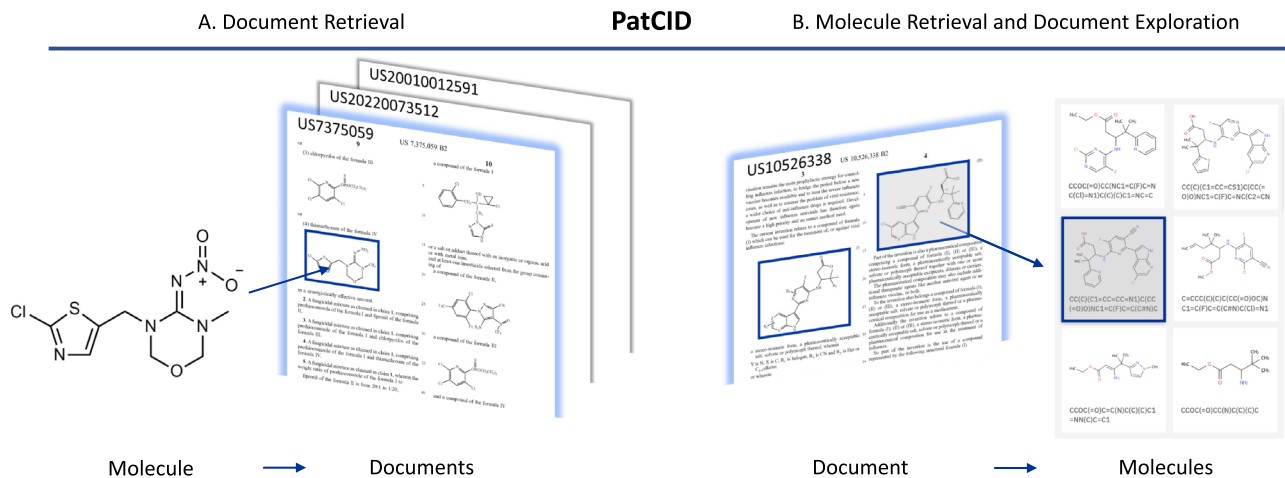

**Fig. 1 | PatCID usage for document and molecule retrieval.** PatCID can **A** retrieve patent documents referring to a query molecule, via similarity, substructure, or as-drawn search. It can also **B** retrieve molecules contained in a document and be used to interactively explore a document collection.

### Table 1 | Patent databases statistics

| Databases | Number of molecules | Number of unique molecules | Number of annotated patent documents | Coverage starting date | | | | |
|---|---|---|---|---|---|---|---|---|
| | | | | U.S. | Europe | Japan | Korea | China |
| **Manual text and image** | | | | | | | | |
| SciFinder[6] | N/A | N/A | N/A | 1973 | 1979 | 1998 | 1995 | 1996 |
| Reaxys[5] | N/A | N/A | N/A | 2001 | 2001 | 2015 | 2015 | 2016 |
| **Automatic, text and image** | | | | | | | | |
| Reaxys | N/A | N/A | N/A | 2000 | 2000 | 2000 | 2000 | 2000 |
| **Automatic, image** | | | | | | | | |
| SureChEMBL[7] | 48.8M | 11.6M | 0.6M | 2007 | 2007 | 2007 | – | – |
| Google Patents | 39.8M | 13.2M | 1.4M | 1911 | 1978 | 2000 | 1998 | 1986 |
| PatCID | 80.7M | 13.8M | 1.2M | 1978 | 1978 | 2004 | 1998 | 2016 |

Number of patent documents, number of molecules, and number of unique molecules covered by patent databases. Which offices are covered and since when.

not allow for effective navigation through large collections and poses a substantial limitation for often lengthy patent documents.

In this work, we present PatCID, the Patent-extracted Chemical-structure Images database for Discovery. PatCID allows users to find patents mentioning a given molecule and, conversely, all molecules covered by specific patents. Leveraging state-of-the-art document understanding models to automatically process patent documents, PatCID bridges the gap between manually- and automatically-created patent chemical-databases. Containing documents from major offices (United States, Europe, Japan, Korea, and China) since 1978, PatCID outperforms other chemical patent databases in terms of coverage and quality for both molecular and document retrieval. PatCID also offers a unique interactive document exploration experience. PatCID accelerates discoveries in chemistry by assisting literature review and by providing a basis for training foundational models in chemistry. The processing pipeline used to build PatCID is published open-source and the dataset, including molecules, document identifiers, and locations is openly accessible[14].

## Results

PatCID is a chemical-structure dataset automatically created from images in patent documents. Figure 1 illustrates its principal usage for document and molecule retrieval. A molecule can be searched with as-drawn, similarity, or substructure search, and a list of patents referencing the molecule is retrieved. On the other hand, molecules selected from a specific document can be extracted and then leveraged to browse and explore the document. PatCID allows persons in the intellectual property domain to carry out prior-art search or landscape

analysis[1], and persons in the organic chemistry domain to review patent literature in various fields such as drug discovery, pharmaceutical chemistry, or material science[15].

To perform a comprehensive evaluation of PatCID, we compare PatCID with state-of-the-art patent databases; the high-level statistics in terms of molecules and documents coverage, the molecular-structure search performances, and the ability to extract molecules from different sections of documents are evaluated. Additionally, we evaluate each component of the processing pipeline used to build PatCID.

### Data statistics

PatCID covers documents from five major patent offices, from the United States (USPTO[16]), Europe (EPO[17]), Japan (JPO[18]), Korea (KIPO[19]), and China (CNIPA[20]). The selected documents are associated with the field of organic chemistry by mentioning the term 'alkyl'. For an exemplary time window of the years 2010–2019, these five patent offices cover 1.06M patent families in the field of organic chemistry, while all 107 patent offices worldwide[21] cover 1.16M, i.e., the offices covered in PatCID represent 90% of published patent documents in the field of organic chemistry. (Here, a patent family refers to the set of patent documents that disclose the same invention, eventually published in different countries.) In total, PatCID indexes 80.7M molecule images, resulting in 13.8M unique chemical structures. This extensive coverage allows the use of PatCID for applications related to various domains of organic chemistry. Additional details related to the collection selection and statistics are provided in Supplementary Note 1.

Table 1 compares key characteristics of state-of-the-art chemical patent databases. It shows the number of patent documents,

molecules, and unique molecules covered by patent databases, as well as which offices are covered and since when. PatCID contains documents that are not manually annotated in Reaxys: the documents published between 1978 and 2001 by the offices in the U.S. and Europe, between 2004 and 2015 in Japan, and between 1998 and 2015 in Korea. PatCID contains 80.7M molecules which is substantially more than Google Patents (39.8M) and SureChEMBL (48.8M). PatCID also contains 13.8M unique molecules, which is more than Google Patents (13.2M) and SureChEMBL (11.6M). Here, molecules (respectively unique molecules) are counted as the number of non-distinct (respectively distinct) canonical Simplified Molecular-Input Line-

Entry System (SMILES)[22] indexed. Additionally, covering Asian Pacific offices is a great advantage over SureChEMBL, as about 70% of patent documents from Asian Pacific offices are not extended to the United States (see Supplementary Note 1). Further information on obtaining the database characteristics can be found in the Method section. For PatCID, detailed statistics by office are also available in Table 2.

## Document ingestion pipeline

PatCID leverages state-of-the-art document understanding models to ingest documents. As illustrated in Fig. 2, the ingestion pipeline uses three components: the document segmentation (DECIMER-Segmentation[23]), the image classification (MolClassifier), and the chemical structure recognition (MolGrapher[24]). The document segmentation module locates the position of chemical images in documents. Chemical images comprise molecular-structure images and Markush-structure[25] images. (Markush structures are sets of molecules defined using positional and frequency variation indications.) To distinguish molecular-structure images and Markush-structure images, we use an image classification module with three output classes: 'Molecular Structure', 'Markush Structure', and 'Background'. This further allows to filter some outliers from the segmentation step, as segmentation errors are included in the 'Background' class. Finally, molecular-structure images are converted to molecular graphs using Mol-Grapher, without stereo-chemistry, and stored as SMILES.

As PatCID is one of the first document-to-molecular-structures pipelines, there is no benchmark for simultaneously evaluating the document segmentation, image classification, and molecule

**Table 2 | PatCID detailed characteristics**

| PatCID coverage | U.S. | Europe | Japan | Korea | China | Total |
|---|---|---|---|---|---|---|
| Since | 1978 | 1978 | 2004 | 1998 | 2016 | – |
| Patent documents processed | 1.2M | 0.5M | 0.8M | 0.3M | 0.2M | 2.8M |
| Pages | 34.7M | 17.8M | 28.6M | 7.4M | 5.0M | 93.5M |
| Images | 80.7M | 23.0M | 30.0M | 10.2M | 7.2M | 151.2M |
| Molecules | 48.2M | 11.2M | 12.9M | 4.7M | 3.8M | 80.7M |

PatCID number of patents processed, number of pages, number of images, and coverage starting date for each patent office. Not all processed patents do necessarily contain annotated chemical structures.

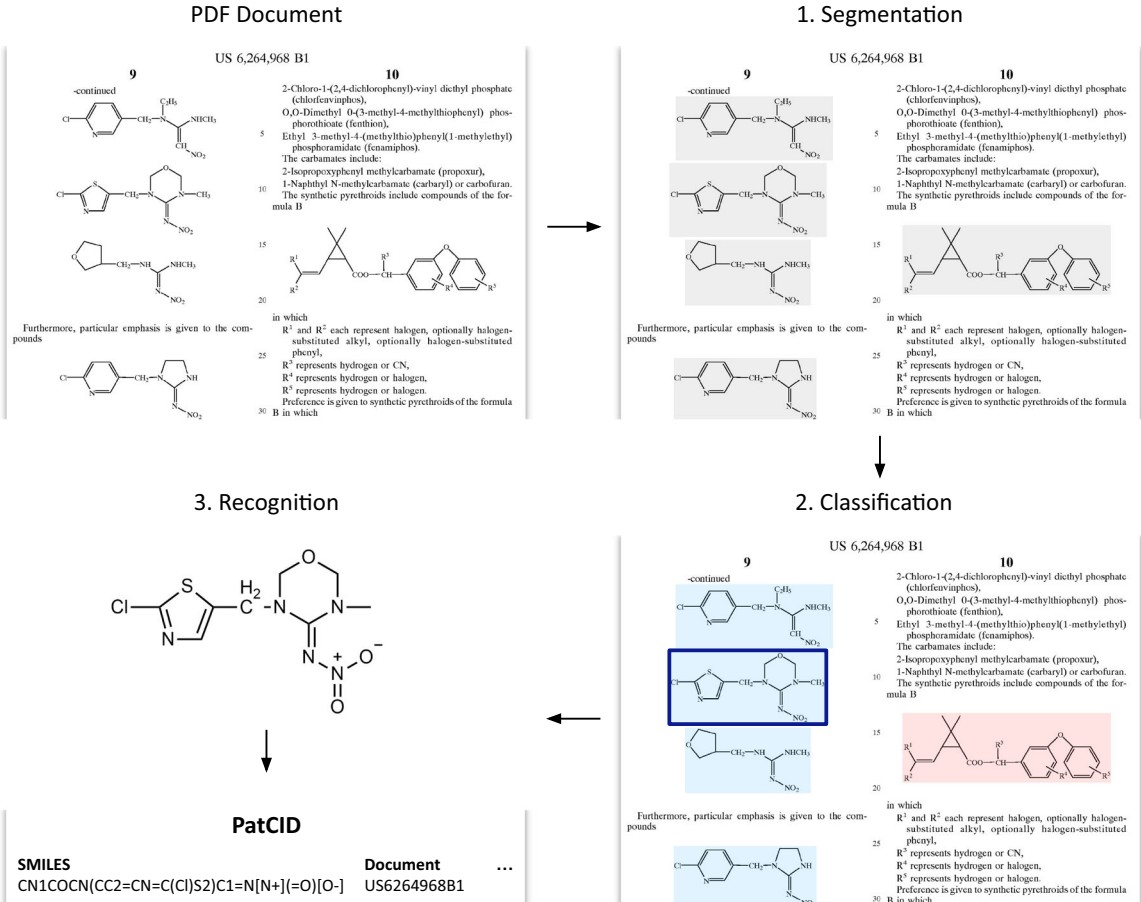

**Fig. 2 | PatCID ingestion pipeline.** The creation of PatCID relies on three key steps: (1) the document page segmentation to extract chemical images, (2) the image classification to identify molecular-structure images, and (3) the molecular recognition to obtain final chemical structures. Blue marks molecular structure images. Red marks Markush structure images.

**Table 3 | Pipeline comparison**

| Components | D2C-RND | | D2C-UNI | |
|---|---|---|---|---|
| | Precision | Recall | Precision | Recall |
| **Segmentation** | | | | |
| YoDe-Segmentation[28] | 45.0% | 42.0% | 37.1% | 35.6% |
| DECIMER-Segmentation[23] | 88.0% | 86.3% | 81.1% | 80.8% |
| **Classification** | | | | |
| MolClassifier | 93.4% | 84.6% | 82.9% | 89.5% |
| **Recognition** | | | | |
| MolScribe[63] | 75.9% | – | 62.7% | – |
| DECIMER[10] | 67.2% | – | 64.6% | – |
| OSRA[64] | 45.6% | – | 41.0% | – |
| MolGrapher | 63.0% (66.3%) | – | 57.1% (61.5%) | – |

Precision and recall of the document ingestion steps: page segmentation, image classification, and chemical-structure recognition. MolGrapher and DECIMER-Segmentation are compared with state-of-the-art models. The numbers in parenthesis are the scores of MolGrapher after filtering the detected errors. Results (in italics) from alternative methods described in the text.

**Table 4 | Search comparison for automatically-created databases**

| Databases | D2C-RND | | D2C-UNI | |
|---|---|---|---|---|
| | Molecules (200) | Annotated Documents (179) | Molecules (164) | Annotated Documents (164) |
| **Automatic, text and image** | | | | |
| SureChEMBL[7] | 23.5% | 45.3% | 6.1% | 52.4% |
| Google patents | 41.5% | 68.2% | 17.7% | 67.1% |
| Reaxys[5] | 41.5% | 59.2% | 36.0% | 58.5% |
| **Automatic, image** | | | | |
| SureChEMBL | 22.0% | 35.8% | 4.9% | 11.6% |
| Google patents | 36.5% | 60.0% | 9.8% | 54.3% |
| PatCID | 56.0% | 100% | 47.6% | 98.2% |

Comparison of the molecule and document retrieval performances of state-of-the-art automatically-created patent-databases. The recall of molecules and annotated documents is reported for benchmarks based on random (D2C-RND) and uniform (D2C-UNI) distributions of chemical images. The numbers in between parentheses are the numbers of samples in each set.

recognition steps. There is even no benchmarks for independently evaluating the document segmentation (with annotated bounding boxes) and the image classification. For this reason, we introduce two benchmark datasets: D2C-RND (Document to Chemical Structures, Random) and D2C-UNI (Document to Chemical Structures, Uniform). Each of these datasets contains three subsets: a first set for evaluating the document segmentation, a second set for image classification, and a third set for the molecule recognition module. Molecules sampled from the recognition subset are taken from images in the classification dataset, which are taken from the pages in the segmentation dataset. This strategy allows us to precisely assess the impact of each module on the overall data quality of the database. D2C-RND is sampled using a random distribution on chemical images, resulting in a higher abundance of recent patents and patents from the U.S. office. This test set can evaluate the average quality of databases. On the other hand, D2C-UNI covers a uniform distribution with respect to the year of publication and publishing office in order to assess databases in challenging scenarios. Specifically, molecule images from older patents and from non-U.S. offices can be of lower quality and use a less standard display style. An example illustrating the diversity of display styles for the same patented molecule in different countries is shown in Supplementary Fig. 6. As the first benchmarks for end-to-end document-to-chemical structures conversion, these benchmarks will benefit future research in this area[14]. In total, they contain 700 manually-annotated pages, 753 manually-annotated chemical images, and 364 precisely annotated molecular graphs (MOL files[26]). More details can be found in the "Methods" section below.

Table 3 presents the performances of these three key ingestion steps. It shows the precision and recall of the page segmentation, the image classification, and the chemical-structure recognition, and for DECIMER-Segmentation and MolGrapher, a comparison with state-of-the-art models. For the recognition module, the precision is computed using InChIKey[27] equality, ignoring stereo-chemistry. The evaluation is performed for the random benchmark D2C-RND and the uniform benchmark D2C-UNI. Further details are available in the "Methods" section.

The segmentation and classification modules achieve high precision and recall of more than 80% on both datasets. The segmentation module outperforms YoDe-Segementation[28] in terms of recall and precision by more than 40% on both benchmarks. The recognition module correctly recognize 63.0% of randomly selected molecule images in PatCID. This is substantially higher than OSRA (45.6%), currently used in automatically-created databases pipelines. On this

dataset DECIMER achieves 67.2% and MolScribe achieves 75.9%. MolScribe was not available at the time PatCID was created. It can also be noted that some images from our benchmarks are part of MolScribe's training data. MolGrapher was preferred over DECIMER for its performance on standard benchmarks (see ref. 24) and its runtime performance advantage, allowing it to be run using CPU only (see Supplementary Table 1). More details on the computational considerations can be found in the Method section below. For all components, models perform better on the random set D2C-RND than on the uniform set D2C-UNI, confirming that documents published recently and in the United States are easier to automatically process. The PatCID ingestion pipeline includes basic filtering steps, such as verifying that the predicted molecular structures contain only one fragment. Based on MolGrapher filtered precision, the precision of the complete PatCID processing pipeline is 54.5% on D2C-RND and 41.3% on D2C-UNI. The recall of the complete pipeline is 46.0% on D2C-RND and 44.5% on D2C-UNI. Qualitative examples of the ingestion pipeline predictions are shown in Supplementary Figs. 1 and 2.

## Search evaluation

In this section, we compare the molecule and document retrieval performance of PatCID with state-of-the-art databases.

Each benchmark dataset contains pairs of molecules and patent documents, from which the molecules have been extracted. By searching for documents in various databases, we compute the document retrieval performance, defined as the percentage of documents retrieved with chemical annotation attached, and we compute the molecule retrieval performance, defined as the percentage of molecules retrieved from the correct reference documents. A query molecule is retrieved if the annotation and ground-truth have identical InChIKeys, ignoring stereo-chemistry. The complete querying process for each database is explained in the Methods section. A comparison of automatically-curated databases will be presented, and a comparison of manually-created databases will follow.

Table 4 compares the recall of molecules and annotated documents of state-of-the-art automatically-created databases on benchmarks D2C-RND and D2C-UNI. For the random set D2C-RND, PatCID achieves a molecule recall of 56.0%, which is higher than Google Patents with visual annotations (36.5%) and higher than Google Patents and Reaxys with visual plus textual annotations (41.5%). For the challenging set D2C-UNI, PatCID achieves a molecule recall of 47.6% and substantially outperforms SureChEMBL with visual annotations (4.9%) and Google Patents with visual annotations (9.8%). It also surpasses

**Table 5 | Search comparison for manually- and automatically-created databases**

| Databases | D2C-RND | | D2C-UNI | |
|---|---|---|---|---|
| | Molecules (200) | Annotated Documents (179) | Molecules (164) | Annotated Documents (164) |
| **Manual, text and image** | | | | |
| SciFinder[6] | 49.5% | N/A | 47.0% | N/A |
| Reaxys[5] | 45.5% | 60.3% | 40.9% | 54.3% |
| **Manual and automatic, text and image** | | | | |
| Reaxys | 53.5% | 68.8% | 51.2% | 67.0% |
| **Automatic, image** | | | | |
| PatCID | 56.0% | 100% | 47.6% | 98.2% |

Comparison of the molecule and document retrieval performances of state-of-the-art manually- and automatically-created patent-databases. The percentage of correctly retrieved molecules and annotated documents is reported for benchmarks based on random (D2C-RND) and uniform (D2C-UNI) distributions of chemical images. The numbers in between parentheses are the numbers of samples in each set.

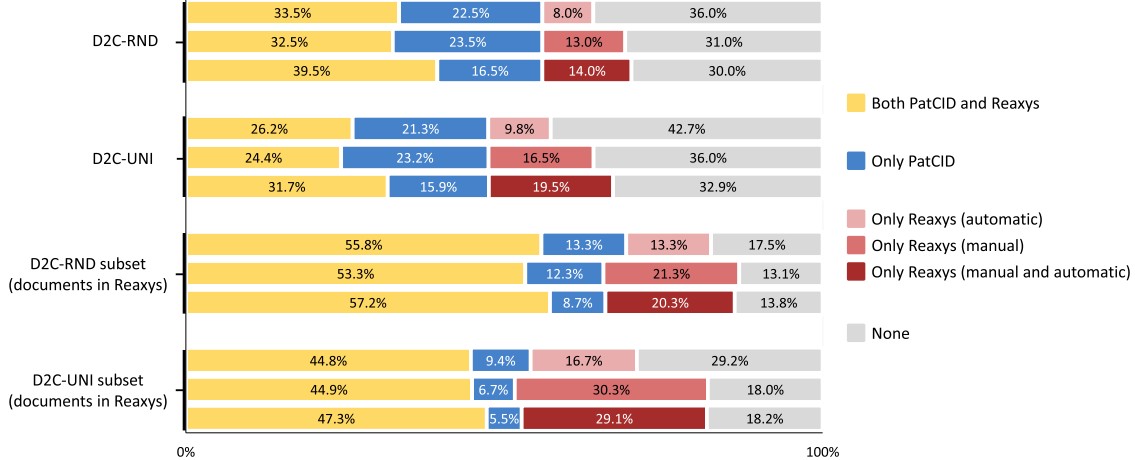

**Fig. 3 | Search comparison between PatCID and Reaxys.** Proportions of molecules covered in the PatCID and Reaxys databases for the random (D2C-RND) and uniform (D2C-UNI) benchmarks, and their subsets restricted to documents annotated in Reaxys. For the top bar, 33.5% of molecules in D2C-RND are covered with both Reaxys (automatic annotations) and PatCID, 22.5% in PatCID only, 8.0% in Reaxys (automatic annotations) only, and 36.0% in none of the databases.

Reaxys with textual and visual annotations by more than 10%. PatCID data quality outperforms all automatic databases by a substantial margin. For D2C-RND, the annotated document recall is 100%, compared to 68.2% in Google Patents, and for D2C-UNI, 98.2%, compared to 67.0% in Google Patents. PatCID has substantially better document coverage. It can be noted that the low document coverage of SureChEMBL is due to the missing coverage of Asian Pacific patent offices. While the PatCID ingestion pipeline only covers visual representation of molecules, its quality and robustness still enable it to surpass state-of-the-art automatically-created databases. SureChEMBL and Google Patents also rely on textual data, and molecular structures information (MOL files) directly provided by the USPTO. Supplementary Table 2 reports the overlap between textual and visual annotations for molecules in Google Patents and SureChEMBL.

Table 5 compares the recall of molecules and annotated documents of state-of-the-art manually- and automatically-created patent databases. PatCID molecule recall outperforms manual annotations of SciFinder for both D2C-RND (56.0% against 49.5%) and D2C-UNI (47.6% against 47.0%). Also, the PatCID annotated document recall is higher than Reaxys with manual and automatic annotations for D2C-RND (100% against 68.8%) and D2C-UNI (98.2% against 67.0%). This advantage of document coverage allows PatCID to compete with Reaxys. Indeed, for D2C-RND, PatCID achieves better molecule retrieval performance than Reaxys, even though Reaxys combines manual and automatic annotations retrieved from images as well as text. Additionally, Reaxys and SciFinder benefit from exploiting the patent families grouping. For example, when searching for a molecule

in a Korean patent, SciFinder and Reaxys are allowed to retrieve the query molecule from any patent in its family, for instance a patent from the U.S. patent office. This is an advantage because the Korean patent depiction style is typically more challenging to automatically process than U.S. patent documents, in which the style is more standardized (see Supplementary Fig. 6). For D2C-UNI, Reaxys has a molecule recall of 51.2%, which is better than the 47.6% molecule recall in PatCID.

Figure 3 illustrates the proportions of molecules covered in the PatCID and Reaxys databases for the random (D2C-RND) and uniform (D2C-UNI) benchmarks, and their subsets restricted to documents annotated in Reaxys. For the D2C-UNI benchmark, Reaxys, with its manual and automatic annotations, covers 51.2% of molecules, while PatCID covers 47.2%, but together they cover a total of 67.1%. Even though Reaxys performs better on average, some of the molecules correctly found in PatCID are not found in Reaxys. Even restricting the evaluation to documents annotated in Reaxys, PatCID covers 8.7% of molecules from D2C-RND and 5.5% of molecules from D2C-UNI, which are not covered in Reaxys. To complement this analysis, a comparison of the number of molecules annotated per patent in PatCID and Reaxys for the random (D2C-RND) benchmark is shown in Supplementary Fig. 5. PatCID bridges the gap between automatically- and manually-created databases, and stands out as a complementary tool to manually-curated databases.

**Document coverage evaluation**

Patent documents in the field of organic chemistry are typically written following two different styles. In the first case, a patent begins by

**Table 6 | Document coverage comparison**

| | Description (before examples) | | Description (examples) | |
|---|---|---|---|---|
| | US20220127225 | US9096558 | US20220127225 | US9096558 |
| Number of molecule images | 1643 | 0 | 138 | 41 |
| Number of molecule images evaluated | 50 | 0 | 50 | 41 |
| SciFinder[6] | 2.0% | – | 94.0% | 100% |
| Reaxys (Manual)[5] | 0% | – | 58.0% | 100% |
| PatCID | 78.0% | – | 82.0% | 100% |

Comparison of the coverage of patent sections in different chemical-structures databases. The percentage of correctly retrieved molecules is reported. Molecules in the description (before examples) section refer to molecules that are displayed and for which no synthesis or properties are provided. Molecules in the description (examples) section refer to molecules that are displayed and for which a synthesis or properties are provided.

**Fig. 4 | PatCID interactive document exploration.** Given a (1) query molecule, PatCID (2) retrieves, as other databases, the reference documents containing the molecule, but additionally (3) retrieves the explicit location of the molecule within documents.

enumerating a large number of molecular structures, and thereafter, for selected key molecules a detailed description and synthetic routes are presented. In the second case, a patent is structured such that from the start, a limited number of molecules is described in detail. Molecules in the description (before examples) section refer to molecules that are displayed and for which no synthesis or properties are provided. Molecules in the description (examples) section refer to molecules that are displayed and for which a synthesis or properties are provided.

This section presents an evaluation of the coverage of different document sections using two documents that are typical examples of different writing styles. US20220127225 has overall a very large number of molecules and for only a few molecules, the synthesis is described in the examples. US9096558 has overall a few molecules, and for all molecules the synthesis is described in the examples. In these two documents, the positions of all chemical structures were manually annotated. In each section, 50 images (if available) were randomly selected and their molecular structures were precisely annotated. In total, this test set was created by manually annotating the position of 1822 molecule images in 235 pages, as well as 141 molecular graphs (MOL files).

Table 6 shows the percentage of correctly retrieved molecules from different patent sections in different chemical-structures databases. PatCID's fully automated process allows it to cover entire documents, including the abstract, the drawings, the description (including examples), and the claims sections. On the other hand, due to the limited workforce, manually-curated databases made the choice to be restricted to the molecule in the examples and to the molecules in the claims section. Doing so, some key patented compounds can be missed. For instance, as illustrated in Table 6, the patent US20220127225 contains mainly molecules before the examples

subsection, with virtually none found in Reaxys or SciFinder, whereas PatCID retrieves 78% of them. An example of a page containing only molecules missed in SciFinder and Reaxys, and almost all found in PatCID, is shown in Supplementary Fig. 3. These molecules illustrated before the examples section can be all the more valuable as some of them are not found in any entries of the entire Reaxys and SciFinder databases. A qualitative example of such molecules is shown in Supplementary Fig. 4. For these reasons, PatCID has a clear advantage over SciFinder and Reaxys with respect to the coverage of sections within documents.

### Interactive document exploration

Figure 4 illustrates an example of document exploration with PatCID. Contrary to SureChEMBL, Google Patents and Reaxys, given a query molecule, PatCID not only finds the documents referencing this molecule but also keeps provenance to its explicit location within documents. For patent documents that can span hundreds of pages and contain thousands of similar molecules, this feature is very useful. It allows to interactively explore documents, easily referring to neighbouring content of the query molecule. It may show related molecular structures or, as depicted in Fig. 4, the synthesis of the molecule.

Providing a dataset of annotated chemical structures, embedded in documents, PatCID can also serve as a foundation for building multimodal document understanding methods[29].

## Discussion

Our extensive comparison between PatCID with state-of-the-art chemical patent-databases shows that recent advances in document mining allow (1) to substantially increase data quality in automatically-created chemical patent-databases and (2) due to better document coverage, to compete with manually-curated databases.

The PatCID ingestion pipeline is based on state-of-the-art document understanding models. Other works introduced workflows for converting PDF documents to chemical structures, including closed-source projects such as MolMiner[30], CLiDE[31], or α-Extractor[32], and the open-source project DECIMER-AI[10]. Similarly to these works, the PatCID ingestion pipeline can process any type of document containing chemical images, such as research articles. However, it is specially optimized for processing patent documents with high precision. Our method also differentiates from others due to its runtime, especially since MolGrapher runs on CPU about 2 times faster than DECIMER-AI (see Supplementary Table 1). The end-to-end document to chemical-structure benchmarks we introduce can also serve as the basis for evaluating future development in this research direction. Reaxys' and SciFinder's manual annotations have an advantage over automatic ingestion pipelines with respect to data quality, as all extracted molecules should be correct. However, for applications such as freedom-to-operate and prior-art search, recall is arguably the most critical metric[1]. This is where PatCID has an advantage. Automatically-generated databases are also claimed to be facing the limitation that key compounds may be hard to find among all annotated compounds, which include solvents, radicals, or fragments[11]. Such irrelevant and abundant molecules would have many occurrences in the database. However, in PatCID, 88% of molecules have less than 5 occurrences, with molecules counted only once per document (see Supplementary Fig. 8). Irrelevant and abundant compounds such as solvents, radicals, or fragments only represent a small fraction of molecules found in PatCID. A comparable analysis leads to the opposite conclusion for SureChEMBL visual and textual annotations[11]. It suggests that such irrelevant compounds are more often found in text, which is not a problem for PatCID, which only takes images into account. Further analysis of the distribution of the number of occurrences of molecules in PatCID is found in Supplementary Note 2. It is worth pointing out that for specific use cases, compound relevancy can be arbitrarily defined, and users may be looking for a way to identify specific subsets of compounds in the database[33–35].

Patent documents contain critical information related to chemical structures, including measured properties, or synthesis paths, which are not necessarily published in research articles[2,3]. To assess the exclusivity of molecules in PatCID, the overlap between molecules in PatCID and PubChem[36] is computed (see the Methods section). Only 7.0M molecules (out of 13.8M) in PatCID are found in PubChem, confirming that PatCID provides novel and exclusive information. A qualitative example of molecules exclusive to PatCID is shown in Supplementary Fig. 4. Additionally, as PatCID contains a large portion of the world's patented molecules, its analysis can provide key elements for understanding patented organic chemistry. Enabling large-scale processing of patent data is not only critical in the patent space. Learning-based molecular generation methods can benefit from PatCID, as a large corpus of 13.8M unique chemical structures can be used for training models[37–39].

In conclusion, PatCID is a chemical-structure database sourced from patent publications. Leveraging state-of-the-art document understanding models, PatCID surpasses automatically-generated databases in terms of data quality by substantial margins. With its extensive document coverage, PatCID can even compete with gold-standard manually-curated databases. PatCID accelerates discoveries in chemistry by assisting with patent literature review, as well as providing a basis for training molecular generation models. In the future, PatCID will also aim to integrate chemical-structures information from text, polymers, and a subset of Markush structures. The processing pipeline used to build PatCID is published open-source, and the dataset, including molecules and their corresponding document locations, is freely accessible for download[14].

## Methods

### Document ingestion pipeline

This section describes the document ingestion pipeline, illustrated in Fig. 2.

**Document segmentation.** For segmentation, the DECIMER-Segmentation model[23] was used. It uses Mask-RCNN[40] to predict an initial mask for each chemical structure in a PDF page and a deterministic mask expansion algorithm to refine the initial masks. The mask expansion algorithm was optimized and multi-processed to allow large-scale processing. The model was trained on pages of patent documents to improve its precision and recall for this application domain. For further details, we refer to publication[23].

**Image classification.** To classify segmented images, we introduce MolClassifier. The image classification module enables Markush structures to be filtered out, since, although recent attempts have been made[41], there is as yet no reliable approach available for the automated recognition of Markush structures at scale. MolClassifier uses Mask R-CNN[40] with three output classes: 'Molecular Structure', 'Markush Structure', and 'Background'. Here, using a segmentation network instead of a classification-only network allows the training to benefit from stronger supervision. Especially, small details at the image level, such as the R-groups are critical to distinguish molecular structures and Markush structures. In this case, label annotations can be converted to mask annotations with no additional cost, given that the images are black and white, and contain only the molecules. An alternative approach can be to classify multiple patches of the chemical image[42]. To train the MolClassifier, a dataset of 15,720 manually-labelled chemical images was created. Selected chemical images are randomly selected from the outputs of the segmentation module for documents from the USPTO. As the first classification dataset for molecules and Markush structures in patent documents, this set can aid future research in this domain[14]. The training images are augmented using standard image augmentations of scaling, rotation, blurring, and noising with pepper patches. Separating the image segmentation and classification modules decomposes the molecule segmentation problem into two simpler tasks. This classification step is particularly easy to train and supervise, using label annotations only.

**Molecule recognition.** The molecule recognition step is performed using MolGrapher[24]. MolGrapher is a graph-based model for converting 2D molecular structure images to machine-readable molecular descriptions. The model comprises a deep keypoint detector and a graph neural network that classifies atoms and bonds. The model demonstrates a precision advantage over rules-based models and is competitive with other learning-based approaches. The model is trained on synthetic images generated using RDKit[43]. Further details can be found in the original publication[24]. This includes an extensive evaluation of standard benchmarks and an analysis of the model robustness. The model robustness is especially important for low-resolution and unconventional images frequently found in documents from patent offices in Asian Pacific. Besides, running MolGrapher at scale allowed us to compute the distribution of common superatoms, i.e. abbreviated substructures, in PatCID (see Supplementary Fig. 9). Such information is valuable to guide future developments of Optical Chemical Structure Recognition models.

### Database evaluation

**Benchmarks and metrics.** Here, we characterize benchmarks and metrics used in the evaluation of the ingestion pipeline and the database.

Two benchmark datasets are introduced: D2C-RND (Document to Chemical structures, Random) and D2C-UNI (Document to Chemical

structures, Uniform). D2C-RND contains 325 pages, 378 images, and 200 molecules, following a random distribution of chemical images. This dataset is intended to reflect the average quality of annotations in PatCID. Since the number of published patents has increased over time, this set contains mainly recent patents. D2C-UNI contains 375 pages, 375 images, and 164 molecules, following a uniform distribution over the year of publication and publishing office of chemical images. This dataset is intended to cover diverse molecules to assess the quality of annotations in a challenging scenario. Older patent documents and from non-U.S. offices generally display molecule images in a less standardized way or with lower resolution and are ultimately more difficult to automatically process with high accuracy. To create these two sets, an intermediate first set was selected by sampling 1400 random pages for each patent office. In these 7000 pages, the location of 15465 chemical images was annotated, including molecules, Markush structures, or polymers with bounding boxes using Label Studio[44]. For each office, the distribution of the number of images per slice of 10 years was computed. Additionally, the intermediate set can be used to estimate the number of images per office in PatCID by finding the number of images per page in the intermediate set and normalizing it with the number of pages in PatCID. Next, the D2C-UNI dataset was created. One image per unique page was selected in the intermediate set. Given that molecules from the same page are likely to be similar, this strategy increases the diversity of molecules in the set. Then, images were sampled according to the uniform distribution over year slices and offices computed from the intermediate set. Finally, we built the D2C-RND dataset. From the intermediate set, images were randomly sampled following the number of images per office estimated in PatCID. Given the limited number of manual annotations performed, the use of two different sampling strategies for the D2C-RND and D2C-UNI benchmarks allows the evaluation to be overall more representative of the full PatCID dataset.

In both datasets, images are annotated by classifying them using three labels: 'Molecular Structure', 'Markush Structure', and 'Background'. For molecular structures, their molecular graph is annotated using the molecule editor ketcher[45]. An application was built to efficiently carry out these annotations (see Supplementary Fig. 10). Especially it allows to import initial predictions from an Optical Chemical Structure Recognition model and edit them, rather than starting from scratch. Graph reconstruction models such as MolGrapher, which preserves atom locations, allow the annotator to quickly map the molecular graph with the image, and gain efficiency.

To evaluate the ingestion pipeline, D2C-RND and D2C-UNI are leveraged. The precision and recall are computed for each ingestion component. For the segmentation module, a predicted bounding box is considered a true positive if its intersection over union with the ground truth is higher than 95%. For the classification module, the precision and recall of the predicted 'Molecular Structure' class are computed. Finally, for the recognition module, the percentage of recognized molecular images is evaluated. In practice, molecules are considered recognized if the prediction and ground truth have identical InChIKeys, ignoring stereo-chemistry.

To compare the chemical patent databases, the annotated document and molecule recall are computed on D2C-RND and D2C-UNI. For each benchmark, which consists of pairs of molecules and associated patents, the annotated document recall is the percentage of retrieved patents with at least one chemical annotation attached. On the other hand, molecule recall is defined as the percentage of molecules that are retrieved and associated with the correct reference patent.

**Databases querying.** This section describes the querying process of each chemical patent-databases: SciFinder, Reaxys, Google Patents, SureChEMBL, PatCID and finally PubChem. The assessment is based on data available in January 2024.

For SciFinder, substances are manually searched by batches of 25 using the advanced search fields 'InChIKey' and 'Patent identifier'. For each batch, SciFinder retrieves a list of molecules matching any of the query InChIKeys and referenced in any of the query patent identifiers. It can be noted that this batching may induce false positives to the advantage of SciFinder, but also considerably accelerates the querying process. In SciFinder, ions are stored together with their counterion as one unique compound. Then, charged molecules can not be matched using their InChIKey. Therefore, queries of charged molecules are instead done using the SciFinder molecule editor, which allows the import of SMILES strings and, ultimately, correctly retrieves charged molecules. SciFinder does not allow to distinguish molecules extracted from text or images, and the matched patent documents can be any patent from the patent family of the query. For Korean patent documents, the patent identifier is the application number, while for other patent offices, it is the publication number. In each office, to get the year of publication of the oldest annotated patent reported in Table 1, we manually search for the oldest patent available in CAS PatentPak, which has substances attached.

For Reaxys, patent documents are manually searched using the 'Query Builder' capabilities. Retrieved patent documents can be downloaded together with their annotated molecules attached. Manual and automatic annotations are searched separately by batches of 50 samples. For fair evaluation, manual annotations are searched using the search field 'Common patent number'. It allows to match any patent from the patent family of the query. We consider that manually annotating any patent in a family indirectly annotates all of them due to their linking. For automatic annotations, patent documents are searched using their 'Patent number', which only matches the exact query document. In this case, annotations are not shared with all documents in the patent family because the quality of automatic annotations depends on the display style of individual patents. Annotated MOL files are obtained from XML files downloaded in Reaxys. The molecules stored as IUPAC[46] names are disregarded. Discriminating the manual and automatic annotations in Reaxys allows us to compare the PatCID and Reaxys automatic ingestion pipelines.

Annotations from Google Patents are retrieved using the BigQuery dataset 'Google Patents Research Data'[47]. The database is queried using the patent publication numbers to get lists of annotated SMILES with sources 'text', 'mol', 'image', or 'pdf' (treated as image). SMILES containing '*' are disregarded, as they define Markush structures and not molecules. In each office, to get the year of publication of the first annotated patent reported in Table 1, patent publication numbers were ordered alphabetically.

For SureChEMBL, the publicly available bulk download[48] of the database is used. For each patent, a list of SMILES is obtained with their sources: 'text', 'mol', or 'image'.

Patent identifier formats are adapted to match each database standard. For Reaxys, Google Patents, and SureChEMBL, we split salts into individual ions to match annotations in our benchmarks. Given that SureChEMBL and Google Patents also rely on MOL files of images directly provided by the USPTO, for a fair comparison, molecules stored in SureChEMBL and Google Patents with a source field 'mol' are counted as images.

For PubChem, the bulk download of SMILES[49] in the database is used. To compute the overlap between PubChem and PatCID efficiently, databases are partitioned into batches where each SMILES is assigned a batch given its number of carbon, nitrogen and oxygen atoms. Then, we compute the percentage of SMILES from PatCID found in PubChem, checking for equality of canonical SMILES strings. We ensure this comparison is valid by computing PubChem and PatCID canonical SMILES using the same RDKit algorithm.

Finally, PatCID stores molecules obtained by running MolGrapher on all segmented chemical images, ignoring the predictions

from the classification module. This allows to maximize the recall of the pipeline, true negatives of MolClassifier still being annotated by MolGrapher.

## Computational considerations

Each processing step is containerized, i.e., the segmentation, classification, and recognition modules. The OpenShift Container Platform is then used to allocate resources at scale. The ingestion is achieved using CPU-only nodes with AMD EPYC 7513 32-Core Processor @2600.000 MHz and 528 GB of RAM. On each CPU node, 32 pods are instanced, each running 4 threads. Using one pod of 4 threads, the average segmentation speed is 8.0 s per page, and the molecule recognition speed is 12.8 s per image. This measure is computed on the USPTO document collection. Optimal resource allocation is achieved by minimizing the number of threads and adhering to memory constraints.

## Reporting summary

Further information on research design is available in the Nature Portfolio Reporting Summary linked to this article.

## Data availability

The PatCID dataset is available on Zenodo[50]. The benchmark datasets are available on Zenodo[51]. The training datasets are available: the image classification dataset can be downloaded on Zenodo[52]; the molecule recognition model training dataset can be downloaded on Hugging Face[53]. The models weights used in this study are available: the document segmentation model, DECIMER-Segmentation[54]; the image classification model, MolClassifier[55]; the molecule recognition model, MolGrapher[56]. To help visualizing the PatCID dataset for test purposes, readers are provided access to a user interface that is currently deployed on IBM's systems by contacting IBM's Deep Search team at deepsearch-core@zurich.ibm.com. Source data are provided with this paper.

## Code availability

Examples showing how to use the PatCID dataset to retrieve molecular structures or patent documents are available on GitHub[14] and Zenodo[57]. The code for the document segmentation model, DECIMER-Segmentation, is available on GitHub[58]. The code for the image classification model, MolClassifier, is available on GitHub[55] and Zenodo[59]. The code for the molecule recognition model, MolGrapher, is available on GitHub[56] and Zenodo[60]. The code for the molecular graph annotation tool is available on GitHub[61] and Zenodo[62].

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

## Acknowledgements

The authors thank Dr. Otto Brinkhaus, Dr. Kohulan Rajan, and Prof. Dr. Christoph Steinbeck for their fruitful interactions.

## Author contributions

L.M., G.I.M. and V.W. conceived the document ingestion pipeline and annotated benchmarks. L.M. performed the database evaluation. P.W.J.S. and F.Y. supervised the work. L.M. wrote the first draft. All authors revised and commented on the manuscript.

## Competing interests

The authors declare no competing interests.
