## [Peer Review File · Nature Communications]

PatCID: an open-access dataset of chemical structures in patent documentsREVIEWER COMMENTS

Reviewer #1 (Remarks to the Author):

Review

PatCID: an open-access dataset of chemical structures in patent documents
Lucas Morin, Valéry Weber, Gerhard Ingmar Meijer, Fisher Yu, Peter W. J. Staar
Nature Communications, NCOMMS-24-09985

Comments to the Editor / Author(s)

In this paper, the authors present a new method for extracting chemical substances from graphical representations in patent documents in detail. Comparing the results obtained with those from existing databases demonstrates the progress made. The paper should, therefore, be published. However, I suggest some corrections and additions listed below in order of importance.

Major issues

1. Collection and document coverage of PatCID in comparison to SciFinder and Reaxys
Lines 55-56, 69, 83-94, 165-166, 248-249; Table 1

The authors repeatedly emphasise that PatCID has better temporal coverage of indexed patent documents than SciFinder and Reaxys.

This is not true for SciFinder:

<https://www.cas.org/support/documentation/references/patyear>

The data shows that SciFinder has better coverage than PatCID for US, JP, KR and CN. The coverage is the same for EP.

The first EP document was published in 1978. This should also be the starting date for PatCID's EP coverage (not 1977).

Table 1 contains the correct coverage starting dates in Reaxys for the manual and automatic annotations. In lines 85-87, the PatCID and Reaxys entry dates are compared so that only the start years of the manual annotation in Reaxys are considered. The start of automatic annotations in 2000 should also be addressed here.

2. PatCID coverage

Lines 76 – 82, Supplementary Note 1, Supplementary Table 1

The data statistics section does not clarify by which procedures patent documents were selected to be processed for PatCID. Possibly, the +alkyl+ keyword filter, as described in Supplementary Note 1, was used; however, this should be made clear.

Within Supplementary Table 1, "Patents Processed" should be substituted by "Patent Documents Processed" if not only granted patents but also patent applications were processed.

Lines 77-80

Specify "time window": The search query in the supplementary note indicates that the publication date is meant.

Furthermore, as specified in the search query (PD:[2010-01-01 TO 2019-12-31], patent families with the publication year 2020 are not included. This should be made clear: publication years are from 2010 to 2019.

Patent families in organic chemistry can neither be precise nor complete be searched using the term +alkyl+ within full texts. A more target-orientated approach uses the truncated patent class C07+ (organic chemistry) (IPC and CPC) instead of the keyword.

3. Search comparison between PatCID and Reaxys

Line 176-183; Figure 3; Lines 346-350

The authors divide Reaxys search results into manually, automatically, and combined manually/automatically derived subsets. The Methods section outlines that this can be achieved using different search fields for patent numbers: "common patent number" vs. "patent number." The different effects of these search fields are correctly explained; however, there is no explanation regarding discrimination between manually and automatically derived subsets in such a way.

Calculation error:

"For the D2C-UNI benchmark, Reaxys, with its manual and automatic annotations, covers 51.2% of molecules, while PatCID covers 47.6% (instead of 47.2%), but together they cover a total of 67.1%."

4. Description section in patent documents

Lines 188-191; Table 5

Patent documents have a standardised format worldwide, comprising bibliographic data, an abstract, a description and the patent claims. As a rule, examples of embodiments can be found at the end of the description. However, this description section is not referred to as the experimental section but simply as examples.

The US20220127225 and US9096558 have both molecule images in the description (including examples), abstracts, and claims. It should be made clear which parts of the patent documents have the number of molecule images in column 1. For US9096558, this number is definitely higher than zero.

5. Less irrelevant compounds in PatCID

Lines 235-237

In SureChEMBL and Google Patents, irrelevant compounds such as solvents are usually extracted from text bodies. This is not an issue for PatCID because only molecule images are considered here.

6. PatCID molecules found in PubChem

Lines 239-240

The cited paper by Kim does not specify the percentage of PatCID molecules found in PubChem (51%). Please explain the origin of this figure.

Additional issues

7. "Patent document" instead of "Patent"

Line 30

Use "patent documents" instead of "patents." The term "patent documents" includes patent applications and (granted) patents.

8. Missing Patentscope

Lines 42-44

Besides SureChEMBL, Google Patents and IBM SIIP Patentscope should be mentioned.

9. Replacing a citation

Lines 45-46, Citation [11]

Concerning SureChEMBL (and Patentscope), deficits in processing quality are better addressed in Ohms, J. Validity of PubChem compounds supplied by Patentscope or SureChEMBL, World Patent Information 70, 102134 (2022)

instead of

Ohms, J. Current methodologies for chemical compound searching in patents: A case study. World Patent Information 66, 102055 (2021).

10. Five major patent offices

Lines 76-77

Korea (KIPO [16]); [16] <https://www.kipo.go.kr/>

Since the home pages of the four other patent offices are given, this should also be the case for Korea. KIPRIS is the website for the Korea Intellectual Property Rights Information Service.

Reviewer #2 (Remarks to the Author):

Morin et al describe PatCID, an open access dataset of chemical structures in patent documents

(although in places a pipeline is mentioned, the title indicates dataset and I have reviewed this article assuming it is about the dataset not a pipeline). The dataset is provided as a 5 GB download without any context prior to download in the figshare repository. Upon download the file contains a minimal readme with the schema of the 6 jsonl files provided. This is not particularly useful to anyone who would want to quickly explore the patent content (there is no web interface or anything similar provided, unlike the other resources they compare their dataset with). The code is present and seems to be open but is distributed amongst several repositories. Bits and pieces of the code clearly build on efforts by other authors (and is credited as such in the article) as well as other efforts from the authors. I find it difficult to know what is new content in this article vs what was previously published. I also do not know who the intended target audience is or what one is meant to use this dataset PatCID for. An evaluation is performed by the authors which claims superior performance to all other databases in the comparison set, yet it is impossible to verify any of these numbers or claims with a reasonable amount of effort and I remain unconvinced. The supporting information is skeletal and the evaluation, while presenting numbers, is not detailed. Quality of structures in patent extraction can be measured in several ways and I do not see any statement about the quality of the chemistry (just many claims that they outperform both manually curated and automated extraction methods). Overall, I see no evidence to refute their numbers or claims, yet I remain completely unconvinced about the quality of this dataset vs others after reading the manuscript, as well as the argumentation applied by the authors, and I do not really know who the intended audience is or what problems the authors wish to solve with this dataset. I was very excited to read this paper at first but now I do not know what I am meant to do with this dataset, should I want to use it. Detailed comments below.

Line 19: 14M unique chemical structures – how was uniqueness determined? Please add a few words here.

L21-2: it is not clear yet what “automatically-created databases” or “manually-curated databases” are.

L23: calling the data “high-quality” doesn’t necessarily prove it. How is the performance quality defined?

L26-7: “the dataset is freely available to download” – where?

L45-6: “Currently these databases fall short in comparison to manually processed databases, both in terms of document coverage and processing quality [10, 11]”. The authors state “currently”, yet the references are from 2015 and 2021, whereas the reference to the dataset in the previous sentence is from 2022. So how was this 2022 dataset included in the comparisons in references 10 and 11? This is not possible. For a fast-moving field with many recent developments, a 2015 reference seems quite out of date.

L46-7: “especially documents published before 2000 or from Asian Pacific patent offices are usually not covered” – usually not? Or are not? If you will compare databases, make sure you know what is in them, don’t speculate.

L49-50: “These documents then need to be manually opened and read” – but the previous sentence describes both automated and manual processes. Surely manual opening and reading of the document is not necessary for automated approaches?

L52: “PatCID” – the name of the dataset – is confusing for anyone familiar with the PubChem CID. I would automatically read it as “Patent Compound ID”.

L53: add comma after “documents” at the end of the line

L55: “major offices” – which major offices?

L55: outperforms (add s to outperform)

L59-60: Here it’s stated that everything is open, but there is no availability statement, or hint where to find this information. I know there may be a statement at the end, but cross-referencing is useful.

Figure 1 looks amazing. Yet if I download the database, I get jsonl files and no images. It feels like there’s a mismatch between what is presented and what is available to the readers in the end.

L61: results before methods – perhaps this is a journal thing, but I find this quite illogical in this case and would prefer the methods first. As obvious from my comments on the results section below, a number of questions arise when reading the results that could/should have been clarified in the methods first and may have made more sense if the methods were earlier.

L67-69: These lines tell the readers how great this is without presenting any results yet. Such statements can only be made after the results are shown, not before.

L78: What are the “patent families” – can these be defined for the purpose of the manuscript?

L80: 90% of published patents – why not 100 %? What about the missing offices?

L88: "Substantially more molecules" – how was this counted?

L89: "more unique molecules" – what is the difference in this sentence and the previous one? Molecules vs unique molecules? One is given in % and one is absolute, this makes them difficult to compare. How was uniqueness defined?

L93: Supplementary Table 1: this is a tiny table and could be included in the main manuscript.

L107: "one of the first document to molecular structures pipeline" – really? But this paper describes a database and not a pipeline? The authors use this as an excuse for no benchmarking precedents, but things could be benchmarked bit-by-bit to be comparable. As is, I find it hard to believe the benchmarking done.

L118: "assess databases in challenging scenarios" – the authors may wish to describe why in more detail for readers less familiar with patent data (e.g. the poorer quality of data in earlier years).

L119-120: "these benchmarks will benefit future research" – but where to access them?

L133: "its runtime performance advantage" – please back up such claims with actual numbers, what was the runtime advantage?

Note: several language issues were noted in the above text (and in other locations) but have not been documented in this review as per the reviewer instructions.

L147-161: While I can't pinpoint exact issues, this evaluation comes across as one-sided to me. To me, it reads as if statistics have been done to show how great PatCID is and any that show the contrary have been omitted. Can this be shown from another perspective?

Figure 3: The dark red category has "Reaxys (manual and automatic) only" – is the "only" necessary? Does the only refer to Reaxys or "manual and automatic"?

L166: "This advantage of document coverage allows PatCID to compete with gold-standard manually curated databases" – in terms of what?

L164-175 General comment: patent retrieval is very noisy, how much of the difference is poor quality noise? I do not see any way to determine this from the evaluation done.

L173-5: What is the outcome of this last sentence? Is Reaxys better here?

L179-183: Again, what are the compounds in this coverage? What is the difference, what is the quality of the compounds that are different?

L186: the use of the word "teaching" seems out of place here?

L204-206: The example with US20220127225 is highlighted to say "these molecules can be critically valuable" – indeed, but they can also be completely meaningless in some contexts. If Reaxys and SciFinder excluded them, perhaps they had reasons? This gets back to my points about the target audience of this collection – it is not clear what / who this resource is designed for. Interestingly, searching for this number in a generic search engine comes up empty – I do not find any information on this patent, so I am unable to comment on the usefulness or not of the data presented in Table 5.

L207-8: "clear advantage ... with respect to coverage" – perhaps, but this depends on the target audience.

L211: which "automatically-created databases" are meant here?

L213: critical – for which purpose?

L219: "our extensive comparison" – as commented above, it does not come across as convincing to me.

L226-227: "pipeline precision and runtime performance" – where are these numbers given?

L230: "perfectly correct" – please define "perfection"?

L231: "the recall is arguably the most critical metric" – this depends on the purpose of the database. I feel the comparison is not very fair and perhaps thus unconvincing since it seems to me that all databases have different aims (are apples being compared with oranges)?

L234: "many duplicates in the database" – why? Isn't the database deduplicated?

L235: "88 % of molecules have less than 5 occurrences" – but how does this happen? What kind of duplicate do you mean here?

L235-7: "This analysis suggests ..." – why don't you do a formal analysis on this? This is just supposition, yet it is possible to classify molecules into their use class (e.g. as solvents) and interrogate your dataset in more detail to support (or deny) this claim. The numbers do not seem to reality-check to me – if most molecules have less than 5 occurrences (assuming a multiple occurrence = multiple mentions in a patent, since it is not clear to me what these duplicates are) – and I know that many, many molecules have thousands and thousands of patent mentions, this seems to indicate a very long tail of very unknown molecules. Are these extraction artefacts? Or genuine molecules?

L239-240: "Interestingly, only 51 % of molecules in PatCID are found in PubChem" – a date of this

query would be interesting, as well as an absolute number behind the 51 % (is this 10, 100, 100 million missing entries?). PubChem do not add all molecules into their database, so it would be interesting to know what is missing? What do they look like? Are there any examples and should they be in PubChem or not?

L240: PatCID provides novel and exclusive information – perhaps yes, but what is the target audience and the quality of this information? Should this be in PubChem or ChemSpider?

L248: I do not feel I've seen sufficient evidence on data quality.

L254: no browsable interface? No access details?

L255 Methods – as commented above, I feel this should be above the results, not after the conclusions.

L262: This (finally) makes it clear that much of this was based on DECIMER, which is one of the leading methods (although with limitations as well).

L325: "molecules are considered recognized of the prediction and ground truth have identical InChIKeys, ignoring stereochemistry" – this is an interesting choice, the lack of handling of stereochemistry should have been obvious much earlier. So was this just an InChIKey first block (skeleton) comparison? What happens if stereochemistry is considered instead? This could result in quite different numbers.

L336/7 (and other locations): I presume the authors have used capitalization from the resource, but the official term is InChIKey according to the InChI Trust.

L340: does this issue with the charged molecules affect the outcomes in any way?

L360-262: "For fair comparison ... MOL files are counted as images" – I don't quite understand this assumption. Extracting data from a MOL file is not the same as extracting from an image – how is this comparing image extraction?

Overall, by the end of the methods, it is clear the authors seem to have done a lot of work, but I remain utterly confused.

L376-379: These lines contain statements where the data is available, without information where it is available.

References: There is an issue with the capitalization of titles, software etc in many of the references.

Supplementary material:

"Supplementary Note 1" – this is so small that it could be included in the methods.

Reviewer #2 (Remarks on code availability):

I reviewed the code availability and links for validity; detailed comments are included in the report above. It is not in a state to be "installed and run" within a reasonable timeframe of review.

Reviewer #3 (Remarks to the Author):

This study is aimed at solving the problem of chemical information retrieval by automatically identifying chemical structures in the patent documents and organize them into database. The main technique proposed is a deep-learning pipeline called PatCID, which can extract molecular structures from multiple chemical patents to construct chemical structure database. The paper reported a newly generated database which is competitive against manually curated databases. The PatCID-created database covers structures from five major patent offices and can be potentially useful to researchers and practitioners in chemistry.

A limitation is that this method could not parse Markush images, table structures, or plain texts, which may restrict the scope and diversity of the generated database. Currently, there has been published works that are capable of recognizing Markush images, like the one published by Wang et al. on Chemical Information Fusion (Briefings in Bioinformatics, 2023).

There are several comments to be considered.

1. Table 2 shows the performance of each step of the pipeline. Could the authors also provide an

overall average precision to further describe the precision of the full recognition pipeline?

2. In line 178-181, the authors mentioned that PatCID could uniquely cover a subset of molecules while Reaxys covers another. Is PatCID having a much better performance in some patents than Reaxys, while not in some other documents? Or for each patent, PatCID and Reaxys both have several unique detections?

3. How to ensure the correctness of the molecule structure in the database constructed from the patents? Is there a filtering step, or the quality control of the database, or things were mainly based on the recognition precision of the pipeline?

4. Can the proposed method be used for scientific papers (e.g. in chemistry or bioinformatics) too? The papers containing structure images may also provide valuable data.

Dear reviewers,

Thank you for your constructive feedback and the suggestions to improve the manuscript "PatCID: an open-access dataset of chemical structures in patent documents" (NCOMMS-24-09985A).

Reviewers' questions and concerns are addressed in detail below. The corresponding modifications in the manuscript are marked **in blue** (in supplementary files).

Reviewer #1

In this paper, the authors present a new method for extracting chemical substances from graphical representations in patent documents in detail. Comparing the results obtained with those from existing databases demonstrates the progress made. The paper should, therefore, be published. However, I suggest some corrections and additions listed below in order of importance.

1. Collection and document coverage of PatCID in comparison to SciFinder and Reaxys

Lines 55-56, 69, 83-94, 165-166, 248-249; Table 1

The authors repeatedly emphasise that PatCID has better temporal coverage of indexed patent documents than SciFinder and Reaxys.

Answer: We agree with reviewer #1 regarding Table 1 and its analysis, and the manuscript was modified accordingly. However, in the introduction and conclusion, the "document coverage" advantage refers to the results from Table 6 (previously Table 5). This table shows that PatCID contains documents which are not annotated in Reaxys and SciFinder (see annotated document recall). This comparison is not limited to the year of publication of the oldest annotated document shown in Table 1.

This is not true for SciFinder: <https://www.cas.org/support/documentation/references/patyear>

The data shows that SciFinder has better coverage than PatCID for US, JP, KR and CN. The coverage is the same for EP.

Answer: Following the reviewer #1 recommendation, we added the temporal coverage of SciFinder to Table 1. We indicated the publication date of the oldest document indexed in CAS PatentPak. The link provided does not precisely indicate if, how, and which part of the document is manually annotated. For example, the starting date provided for the U.S. is 1828, whereas in a manual search in SciFinder, the oldest U.S. patent found with an annotation was published in 1878. Furthermore, between 1878 and 1973, patent documents in SciFinder seem to contain very few molecules. These annotations could come from the document title, or from the abstract only.

The first EP document was published in 1978. This should also be the starting date for PatCID's EP coverage (not 1977).

Answer: We corrected the manuscript accordingly.

Table 1 contains the correct coverage starting dates in Reaxys for the manual and automatic annotations. In lines 85-87, the PatCID and Reaxys entry dates are compared so that only the start years of the manual annotation in Reaxys are considered. The start of automatic annotations in 2000 should also be addressed here.

Answer: We updated the sentence to specifically mention Reaxys manual annotations.

2. PatCID coverage

Lines 76 – 82, Supplementary Note 1, Supplementary Table 1

The data statistics section does not clarify by which procedures patent documents were selected to be processed for PatCID. Possibly, the +alkyl+ keyword filter, as described in Supplementary Note 1, was used; however, this should be made clear.

Answer: As suggested, we added this precision in the “Results - Data Statistics” section of the manuscript.

Within Supplementary Table 1, “Patents Processed” should be substituted by “Patent Documents Processed” if not only granted patents but also patent applications were processed.

Lines 77-80 Specify “time window”: The search query in the supplementary note indicates that the publication date is meant.

Furthermore, as specified in the search query (PD:[2010-01-01 TO 2019-12-31], patent families with the publication year 2020 are not included. This should be made clear: publication years are from 2010 to 2019.

Answer: We modified the manuscript accordingly.

Patent families in organic chemistry can neither be precise nor complete be searched using the term +alkyl+ within full texts. A more target-orientated approach uses the truncated patent class C07+ (organic chemistry) (IPC and CPC) instead of the keyword.

Answer: We thank the reviewer #1 for this suggestion, and added in the Supplementary Note 1 a discussion about the documents selection process. We agree that using organic chemistry IPC codes would be a relevant approach. However, on a set manually examined documents we found that (1) many patent documents containing molecule images are not classified with IPC codes C07, C08 and C09, and (2) a large majority of documents classified as C07, C08, C09 do not contain molecule images. Given constraints of processing cost, the search term “alkyl” was preferred.

3. Search comparison between PatCID and Reaxys

Line 176-183; Figure 3; Lines 346-350

The authors divide Reaxys search results into manually, automatically, and combined manually/automatically derived subsets. The Methods section outlines that this can be achieved using different search fields for patent numbers: “common patent number” vs. “patent number.” The different effects of these search fields are correctly explained; however, there is no explanation regarding discrimination between manually and automatically derived subsets in such a way.

Answer: In Reaxys, querying using the “common patent number” or the “patent number” is independent from filtering automatic or manual annotations. Reaxys is queried in the way described as we consider this makes the evaluation fair. We clarified this point in the “Method - Databases Querying” section. Besides, discriminating the manual and automatic annotations in Reaxys makes it possible to compare both PatCID and Reaxys, as well as their respective automatic ingestion pipelines.

Calculation error:

“For the D2C-UNI benchmark, Reaxys, with its manual and automatic annotations, covers 51.2% of molecules, while PatCID covers 47.6% (instead of 47.2%), but together they cover a total of 67.1%.”

Answer: We modified the manuscript accordingly.

4. Description section in patent documents

Lines 188-191; Table 5

Patent documents have a standardised format worldwide, comprising bibliographic data, an abstract, a description and the patent claims. As a rule, examples of embodiments can be found at the end of the description. However, this description section is not referred to as the experimental section but simply as examples. The US20220127225 and US9096558 have both molecule images in the description (including examples), abstracts, and claims. It should be made clear which parts of the patent documents have the number of molecule images in column 1. For US9096558, this number is definitely higher than zero.

Answer: We agree with reviewer #1 that using the nomenclature “description” to refer to any information before the “examples”, and using “experimental section” to refer to any information in, or after the “examples”,

can be misleading. We replaced in the manuscript the two parts of the documents under consideration by “description (before examples)” and “examples”.

5. Less irrelevant compounds in PatCID

Lines 235-237

In SureChEMBL and Google Patents, irrelevant compounds such as solvents are usually extracted from text bodies. This is not an issue for PatCID because only molecule images are considered here.

Answer: As suggested, we added this clarification to the manuscript.

6. PatCID molecules found in PubChem

Lines 239-240

The cited paper by Kim does not specify the percentage of PatCID molecules found in PubChem (51%). Please explain the origin of this figure.

Answer: We reformulated this sentence and added details on obtaining the PatCID and PubChem overlap in the “Method - Databases Querying” section.

7. Additional issues “Patent document” instead of “Patent”

Line 30

Use “patent documents” instead of “patents.” The term “patent documents” includes patent applications and (granted) patents.

8. Missing Patentscope

Lines 42-44

Besides SureChEMBL, Google Patents and IBM SIIP Patentscope should be mentioned.

9. Replacing a citation

Lines 45-46, Citation [11]

Concerning SureChEMBL (and Patentscope), deficits in processing quality are better addressed in Ohms, J. Validity of PubChem compounds supplied by Patentscope or SureChEMBL, World Patent Information 70, 102134 (2022)

instead of

Ohms, J. Current methodologies for chemical compound searching in patents: A case study. World Patent Information 66, 102055 (2021).

10. Five major patent offices

Lines 76-77

Korea (KIPO [16]); [16] <https://www.kipo.go.kr/>

Since the home pages of the four other patent offices are given, this should also be the case for Korea. KIPRIS is the website for the Korea Intellectual Property Rights Information Service.

Answer: We corrected these points in the manuscript accordingly.

Reviewer #2

Morin et al describe PatCID, an open access dataset of chemical structures in patent documents (although in places a pipeline is mentioned, the title indicates dataset and I have reviewed this article assuming it is about the dataset not a pipeline). The dataset is provided as a 5 GB download without any context prior to download in the figshare repository. Upon download the file contains a minimal readme with the schema of the 6 json files provided. This is not particularly useful to anyone who would want to quickly explore the patent content (there is no web interface or anything similar provided, unlike the other resources they compare their dataset with).

Answer: We agree with reviewer #2 that improving the accessibility to the dataset is relevant. We now created a GitHub repository with examples showing how to use the PatCID dataset to retrieve molecular structures or

patent documents [1]. Additionally, we created a user interface which is showcased in the same repository. Reviewers are invited to request access by contacting the IBM's Deep Search team at deepsearch-core@zurich.ibm.com.

The code is present and seems to be open but is distributed amongst several repositories. Bits and pieces of the code clearly build on efforts by other authors (and is credited as such in the article) as well as other efforts from the authors. I find it difficult to know what is new content in this article vs what was previously published.

Answer: The new GitHub repository [1] now centralizes all published code, datasets and models.

I also do not know who the intended target audience is or what one is meant to use this dataset PatCID for.

Answer: The target audiences are persons in the intellectual-property domain and persons in the organic chemistry domain. PatCID can be used, for example, to perform a prior-art search to identify patent documents that protect a chemical structure. (Prior-art prevents the use of said chemical structure by others). Alternatively, persons in the organic chemistry domain can use PatCID to review patent literature in various fields such as drug discovery, pharmaceutical chemistry, or material science. In general, PatCID can be used to search for patent documents relating to specific molecules or, conversely, to find which molecules appear in which patents.

An evaluation is performed by the authors which claims superior performance to all other databases in the comparison set, yet it is impossible to verify any of these numbers or claims with a reasonable amount of effort and I remain unconvinced. The supporting information is skeletal and the evaluation, while presenting numbers, is not detailed.

Answer: The examples in the new GitHub repository [1] provide access to PatCID annotations. Google Patents, Reaxys, and SciFinder annotations can not be distributed due to legal constraints.

Quality of structures in patent extraction can be measured in several ways and I do not see any statement about the quality of the chemistry (just many claims that they outperform both manually curated and automated extraction methods).

Answer: If the quality of the chemistry refers to the quality of PatCID annotations compared to reference molecules displayed in patents, then we measure quality using InChIKey equality, ignoring stereochemistry. This clarification has been added in the "Results" section. If the quality of the chemistry is taken to refer to the relevance of the molecules originally found in patent documents, we would like to emphasize that PatCID simply aims to include all chemical structures found in patent document images. For specific use cases, the relevance of compounds can be arbitrarily defined, but our evaluation is not biased towards a specific application domain.

Overall, I see no evidence to refute their numbers or claims, yet I remain completely unconvinced about the quality of this dataset vs others after reading the manuscript, as well as the argumentation applied by the authors, and I do not really know who the intended audience is or what problems the authors wish to solve with this dataset. I was very excited to read this paper at first but now I do not know what I am meant to do with this dataset, should I want to use it. Detailed comments below

Line 19: 14M unique chemical structures – how was uniqueness determined? Please add a few words here.

Answer: The number of unique molecule is the number of distinct canonical SMILES. As suggested, we clarified this point in the "Results - Data Statistics" section, because the number of words in the abstract word is limited.

L21-2: it is not clear yet what "automatically-created databases" or "manually-curated databases" are.

Answer: Automatically-created databases are Google Patents and SureChEMBL [2], and manually-curated databases are Reaxys [3] and SciFinder [4]. As suggested, we clarified this point in the revised manuscript.

L23: calling the data "high-quality" doesn't necessarily prove it. How is the performance quality defined?

Answer: Here, the performance is defined as the percentage of molecules retrieved in the database.

L26-7: “the dataset is freely available to download” – where?

Answer: As suggested, we added a reference to the new GitHub repository [1].

L45-6: “Currently these databases fall short in comparison to manually processed databases, both in terms of document coverage and processing quality [10, 11]”. The authors state “currently”, yet the references are from 2015 and 2021, whereas the reference to the dataset in the previous sentence is from 2022. So how was this 2022 dataset included in the comparisons in references 10 and 11? This is not possible. For a fast-moving field with many recent developments, a 2015 reference seems quite out of date.

Answer: The citation related to Google Patents and published in 2022 has been removed from the manuscript. Google Patents was launched in 2006. As also suggested by reviewer #1, a more recent article comparing patent databases has been added to the manuscript.

L46-7: “especially documents published before 2000 or from Asian Pacific patent offices are usually not covered” – usually not? Or are not? If you will compare databases, make sure you know what is in them, don’t speculate.

Answer: We corrected this point in the manuscript accordingly.

L49-50: “These documents then need to be manually opened and read” – but the previous sentence describes both automated and manual processes. Surely manual opening and reading of the document is not necessary for automated approaches?

Answer: Databases created automatically or manually, allow retrieving document identifiers from a query molecule. To inspect these documents, users have to open the document and manually search for the pages of interest (mentioning the molecule that they are looking for). PatCID, on the other hand, stores the exact position in the document from which the molecule is found. This makes a difference, as some patents can span hundreds of pages. We clarified this point in the revised manuscript.

L52: “PatCID” – the name of the dataset – is confusing for anyone familiar with the PubChem CID. I would automatically read it as “Patent Compound ID”.

Answer: We hope that the explicit definition of PatCID acronyms given at the beginning of the abstract will resolve any ambiguity.

L53: add comma after “documents” at the end of the line

Answer: We corrected this point in the manuscript accordingly.

L55: “major offices” – which major offices?

Answer: The major offices covered in PatCID are the United States, Europe, Japan, Korea, and China. We modified the manuscript accordingly.

L55: outperforms (add s to outperform)

L59-60: Here it’s stated that everything is open, but there is no availability statement, or hint where to find this information. I know there may be a statement at the end, but cross-referencing is useful.

Answer: We corrected these points in the manuscript accordingly.

Figure 1 looks amazing. Yet if I download the database, I get jsonl files and no images. It feels like there’s a mismatch between what is presented and what is available to the readers in the end.

Answer: We hope that the new GitHub repository [1] with examples showing how to use PatCID to retrieve molecules or documents can answer this concern. In the provided examples, PDF documents need to be downloaded from the official patent offices website due to legal constraints on distribution. If it can be useful for the scientific community, we will distribute all segmented chemical images (3.6 TiB of data), for any patent office that allows it.

L61: results before methods – perhaps this is a journal thing, but I find this quite illogical in this case and would prefer the methods first. As obvious from my comments on the results section below, a number of questions arise when reading the results that could/should have been clarified in the methods first and may have made more sense if the methods were earlier.

Answer: We follow the structure of articles published in Nature Communications. In the revised manuscript, we clarified the “Results” section to make sure that the main ideas are clear without reading the “Methods” section.

L67-69: These lines tell the readers how great this is without presenting any results yet. Such statements can only be made after the results are shown, not before.

Answer: We corrected this point in the manuscript accordingly.

L78: What are the “patent families” – can these be defined for the purpose of the manuscript?

Answer: As suggested, we added this clarification to the revised manuscript.

L80: 90% of published patents – why not 100 %? What about the missing offices?

Answer: Processing documents from all offices would provide 10% more documents, but also require to collect documents for 100 additional offices. Obtaining these documents can be subject to costs for each patent office.

L88: “Substantially more molecules” – how was this counted?

Answer: The number of molecules is the number of non-distinct canonical SMILES. As suggested, we added this clarification to the manuscript.

L89: “more unique molecules” – what is the difference in this sentence and the previous one? Molecules vs unique molecules? One is given in % and one is absolute, this makes them difficult to compare. How was uniqueness defined?

Answer: The number of unique molecules is the number of distinct canonical SMILES. As suggested, we added this clarification to the manuscript.

L93: Supplementary Table 1: this is a tiny table and could be included in the main manuscript.

Answer: As suggested, we included this table in the main manuscript.

L107: “one of the first document to molecular structures pipeline” – really? But this paper describes a database and not a pipeline? The authors use this as an excuse for no benchmarking precedents, but things could be benchmarked bit-by-bit to be comparable. As is, I find it hard to believe the benchmarking done.

Answer: The benchmarks introduced in this article are the only available for evaluating simultaneously all components of a document to chemical-structure pipeline. To complement our evaluation, we added a comparison with YoDe-Segmentation [5] in Table 3 (previously Table 2). ChemSAM [6] would be another alternative but the code is not yet available. For the classification, the only similar work published [7] does not yet provide any model checkpoint. For the recognition, the original publication of MolGrapher [8] contains an extensive evaluation against available models.

L118: “assess databases in challenging scenarios” – the authors may wish to describe why in more detail for readers less familiar with patent data (e.g. the poorer quality of data in earlier years).

Answer: As suggested, we added this clarification to the manuscript. Additionally, an example of the different display styles in patent offices is added as Supplementary Fig. 6.

L119-120: “these benchmarks will benefit future research” – but where to access them?

Answer: We corrected this point in the manuscript accordingly.

L133: “its runtime performance advantage” – please back up such claims with actual numbers, what was the runtime advantage?

Answer: As suggested, we added a comparison of runtimes between DECIMER and MolGrapher in Supplementary Table 1.

Note: several language issues were noted in the above text (and in other locations) but have not been documented in this review as per the reviewer instructions.

L147-161: While I can't pinpoint exact issues, this evaluation comes across as one-sided to me. To me, it reads as if statistics have been done to show how great PatCID is and any that show the contrary have been omitted. Can this be shown from another perspective?

Answer: Our search evaluation is based on two independent strategies to select test documents (randomly or uniformly sampled over years and offices). It is the only systematic evaluation done at this scale for patent databases (manual annotation of 700 pages, 753 chemical images, and 364 molecules).

Figure 3: The dark red category has “Reaxys (manual and automatic) only” – is the “only” necessary? Does the only refer to Reaxys or “manual and automatic”?

Answer: “Only” refers to Reaxys. It indicates the molecules which are found in Reaxys but not in PatCID. For greater clarity, this item has been reworded as “Only Reaxys (manual and automatic)”.

L166: “This advantage of document coverage allows PatCID to compete with gold-standard manually curated databases” – in terms of what?

Answer: Table 5. (previously Table 4.) shows that some documents have molecules attached in PatCID, but not in Reaxys. It is the main reason why PatCID retrieves more molecules than Reaxys. We hope this point is clarified in the revised manuscript.

L164-175 General comment: patent retrieval is very noisy, how much of the difference is poor quality noise? I do not see any way to determine this from the evaluation done.

Answer: We thank reviewer #2 for raising this discussion. PatCID aims to include all chemical structures found in patent documents images. This is the objective assessed using our evaluation framework. In the “Discussion” section, we discuss this, pointing out that patent documents can contain multiple mentions of abundant and irrelevant molecules such as solvents. Senger et al. [9] also raised this question for annotations in SureChEMBL. Following the same methodology, we show that contrary to SureChEMBL, PatCID contains mostly molecules that appear less than five times in different documents. Likely, most of the noisy information from SureChEMBL comes from the text and not images. Besides, in this discussion, relevance is measured using the abundance, but it is worth pointing out that for specific use cases, compound relevancy can be arbitrarily defined. Our evaluation framework is not biased towards a specific application domain. We added these points in the “Discussion” section.

L173-5: What is the outcome of this last sentence? Is Reaxys better here?

Answer: On the D2C-UNI benchmark, Reaxys is better on average. However, as shown in Figure 3, some of the molecules correctly retrieved in PatCID are not in Reaxys. The manuscript was modified in order for this conclusion to come after the analysis of Figure 3.

L179-183: Again, what are the compounds in this coverage? What is the difference, what is the quality of the compounds that are different?

Answer: To illustrate this point, we added in Supplementary Fig. 4 a qualitative example of a page with molecules found in PatCID, but not in Reaxys, SciFinder or PubChem.

L186: the use of the word “teaching” seems out of place here?

Answer: As suggested, we modified this point in the revised manuscript.

L204-206: The example with US20220127225 is highlighted to say “these molecules can be critically valuable” – indeed, but they can also be completely meaningless in some contexts. If Reaxys and SciFinder excluded them, perhaps they had reasons? This gets back to my points about the target audience of this collection – it is not clear what / who this resource is designed for. Interestingly, searching for this number in a generic search engine comes up empty – I do not find any information on this patent, so I am unable to comment on the usefulness or not of the data presented in Table 5.

Answer: SciFinder and Reaxys are limited by the manual involvement in their annotation pipeline [9] which restrict their manual processing to specific document sections. To better illustrate this point, we added in the Supplementary Fig. 3 an example of a page from US20220127225 containing only molecules missed in SciFinder and Reaxys, but almost all found in PatCID. The patents US20220127225 and US9096558 used for the document coverage evaluation are accessible in SciFinder (References search: US20220127225, US9096558), in Reaxys (Quick search: US20220127225, US9096558), and in Google Patents [10] [11].

L207-8: “clear advantage ... with respect to coverage” – perhaps, but this depends on the target audience.

Answer: By covering more sections of the document, more molecules can be captured. Capturing a larger number of molecules is valuable, for instance to perform a prior-art search [12].

L211: which “automatically-created databases” are meant here?

Answer: As suggested, we clarified this point in the revised manuscript.

L213: critical – for which purpose?

Answer: As suggested, we clarified this point in the revised manuscript.

L219: “our extensive comparison” – as commented above, it does not come across as convincing to me.

Answer: We compare patent databases statistics, search performances and document section coverage. Our search evaluation is based on two independent strategies to select test documents (randomly or uniformly sampled over years and offices). It is the only systematic evaluation done at this scale for all patent databases (manual annotation of 700 pages, 753 chemical images, and 364 molecules).

L226-227: “pipeline precision and runtime performance” – where are these numbers given?

Answer: As suggested, we added a comparison of runtimes between DECIMER and MolGrapher in Supplementary Table 1.

L230: “perfectly correct” – please define “perfection”?

Answer: As suggested, we added this clarification to the manuscript.

L231: “the recall is arguably the most critical metric” – this depends on the purpose of the database.

Answer: The purpose of the database is discussed above. For prior-art search, the recall is the most important metric [12].

I feel the comparison is not very fair and perhaps thus unconvincing since it seems to me that all databases have different aims (are apples being compared with oranges)?

Answer: SureChEMBL, Google Patents, Reaxys and SciFinder are also tools for retrieval of information about chemical compounds in patents [13, 14]. Although the search criteria may be different, these databases, as PatCID, can be used to retrieve source documents from query molecules. We propose a fair evaluation of these molecular and document retrieval capabilities.

L234: “many duplicates in the database” – why? Isn’t the database deduplicated?

Answer: The same molecule may appear several times in one document, but also several times in several documents. For each occurrence, the database stores one entry, with the same SMILES but a different source location. This allows access to the mapping between a molecule and all its references.

L235: “88 % of molecules have less than 5 occurrences” – but how does this happen? What kind of duplicate do you mean here?

Answer: Duplicates are molecules having the same canonical SMILES. 88% of molecules have less than 5 occurrences from different patent documents, since most molecules are only found in documents from the same patent family. (In general, there should be at most 5 patents in the same family, since 5 patent offices were processed.)

L235-7: “This analysis suggests ...” – why don’t you do a formal analysis on this? This is just supposition, yet it is possible to classify molecules into their use class (e.g. as solvents) and interrogate your dataset in more detail to support (or deny) this claim.

Answer: From the molecule image, the use of the molecule can not be easily determined. For instance, a reactant can be a product or a solvent, depending on the context.

The numbers do not seem to reality-check to me – if most molecules have less than 5 occurrences (assuming a multiple occurrence = multiple mentions in a patent, since it is not clear to me what these duplicates are) – and I know that many, many molecules have thousands and thousands of patent mentions, this seems to indicate a very long tail of very unknown molecules. Are these extraction artefacts? Or genuine molecules?

Answer: We agree with reviewer #2 that analyzing this point is relevant. We added as Supplementary Fig. 7 illustrations of molecules highly abundant in PatCID (Markush structures R-group examples, reactants, and counterions). We also added in Supplementary Note 2, an evaluation of 231 molecules which appear only once or twice in the database. The result is consistent with the evaluation on the random benchmark (D2C-RND), and disproves the idea that molecules with low number of occurrences are necessarily errors.

L239-240: “Interestingly, only 51 % of molecules in PatCID are found in PubChem” – a date of this query would be interesting, as well as an absolute number behind the 51 % (is this 10, 100, 100 million missing entries?).

Answer: As suggested, we added this clarification to the manuscript.

PubChem do not add all molecules into their database, so it would be interesting to know what is missing? What do they look like? Are there any examples and should they be in PubChem or not?

Answer: To illustrate this point, we added in Supplementary Fig. 4 a qualitative example of a page with molecules found in PatCID, but not in PubChem.

L240: PatCID provides novel and exclusive information – perhaps yes, but what is the target audience and the quality of this information? Should this be in PubChem or ChemSpider?

Answer: We thank referee #2 for this suggestion, and will consider it in the future.

L248: I do not feel I’ve seen sufficient evidence on data quality.

Answer: Our search evaluation is based on two independent strategies to select test documents (randomly or uniformly sampled over years and offices). It is the only systematic evaluation done at this scale for all patent databases (manual annotation of 700 pages, 753 chemical images, and 364 molecules).

L254: no browsable interface? No access details?

Answer: We created a user interface which is showcased in the new GitHub repository [1]. Reviewers are invited to request access by contacting the IBM's Deep Search team at deepsearch-core@zurich.ibm.com.

L255 Methods – as commented above, I feel this should be above the results, not after the conclusions.

Answer: We follow the structure of articles published in Nature Communications.

L262: This (finally) makes it clear that much of this was based on DECIMER, which is one of the leading methods (although with limitations as well).

Answer: One of the three processing modules of the pipeline is DECIMER-Segmentation. This is mentioned when the pipeline is introduced, in the “Result - Document Ingestion Pipeline” section.

L325: “molecules are considered recognized of the prediction and ground truth have identical InChIKeys, ignoring stereochemistry” – this is an interesting choice, the lack of handling of stereochemistry should have been obvious much earlier. So was this just an InChIKey first block (skeleton) comparison? What happens if stereochemistry is considered instead? This could result in quite different numbers.

Answer: As suggested, we added this clarification to the manuscript. It is worth pointing out that ignoring stereo-chemistry does not negatively impact the pipeline recall, which we consider the most important metric for a search application.

L336/7 (and other locations): I presume the authors have used capitalization from the resource, but the official term is InChIKey according to the InChI Trust.

Answer: We corrected this point in the manuscript accordingly.

L340: does this issue with the charged molecules affect the outcomes in any way?

Answer: No, it does not affect the results. We reworded the sentence in the revised manuscript.

L360-262: “For fair comparison ... MOL files are counted as images” – I don’t quite understand this assumption. Extracting data from a MOL file is not the same as extracting from an image – how is this comparing image extraction?

Answer: For documents recently published in the United States, MOL files describing the molecular structures depicted in images are directly provided. SureChEMBL and Google Patents rely on this data. They index these molecules with a source field “mol”, which we treat as a source field ‘image’. This setup can advantage SureChEMBL and Google Patents, but it is fair since these molecules indeed come from molecule images, and SureChEMBL and Google Patents do not necessarily need to process these images again.

Overall, by the end of the methods, it is clear the authors seem to have done a lot of work, but I remain utterly confused.

L376-379: These lines contain statements where the data is available, without information where it is available.

References: There is an issue with the capitalization of titles, software etc in many of the references.

Answer: We corrected this point in the manuscript accordingly.

Supplementary material:

“Supplementary Note 1” – this is so small that it could be included in the methods.

Answer: Following suggestions from reviewer #1, the Supplementary Note 1 was extended. We consider it is now large enough to be in the Supplementary Materials.

Reviewer #2 (Remarks on code availability):

I reviewed the code availability and links for validity; detailed comments are included in the report above. It is not in a state to be "installed and run" within a reasonable timeframe of review.

Answer: The code available allows to search molecules and documents in PatCID, as well as running all models used to build the dataset.

Reviewer #3

This study is aimed at solving the problem of chemical information retrieval by automatically identifying chemical structures in the patent documents and organize them into database. The main technique proposed is a deep-learning pipeline called PatCID, which can extract molecular structures from multiple chemical patents to construct chemical structure database. The paper reported a newly generated database which is competitive against manually curated databases. The PatCID-created database covers structures from five major patent offices and can be potentially useful to researchers and practitioners in chemistry.

A limitation is that this method could not parse Markush images, table structures, or plain texts, which may restrict the scope and diversity of the generated database. Currently, there has been published works that are capable of recognizing Markush images, like the one published by Wang et al. on Chemical Information Fusion (Briefings in Bioinformatics, 2023).

Answer: We agree that automated processing of Markush structures depicted in documents would be valuable. Wang et al. [15] describe the only model for jointly recognizing Markush structure images (with positional and frequency variation indications), and R-groups textual descriptions. However, this work does not discuss the segmentation of visual and textual description of Markush structures, as well as their matching. Due to this manual involvement, this approach cannot be used to automatically process documents at scale. Additionally, the model, code and data used in this study are not available. We added this comment in the “Method - Document Ingestion Pipeline - Image Classification” section of the revised manuscript

1. There are several comments to be considered. Table 2 shows the performance of each step of the pipeline. Could the authors also provide an overall average precision to further describe the precision of the full recognition pipeline?

Answer: Based on MolGrapher filtered precision, the precision of the complete PatCID processing pipeline is 54.5% on the random benchmark (D2C-RND) and 41.3% on the uniform benchmark (D2C-UNI). The recall of the complete pipeline is 46.0% on the random benchmark (D2C-RND) and 44.5% on the uniform benchmark (D2C-UNI). We added this information in the “Results - Document Ingestion Pipeline” section.

2. In line 178-181, the authors mentioned that PatCID could uniquely cover a subset of molecules while Reaxys covers another. Is PatCID having a much better performance in some patents than Reaxys, while not in some other documents? Or for each patent, PatCID and Reaxys both have several unique detections?

Answer: First of all, it can be noted in Table 5 (previously Table 4) that some documents are annotated in PatCID, but not in Reaxys (see annotated document recall). For these, PatCID necessarily performs much better on full documents. To take this a step further, we added as Supplementary Fig. 5 a histogram of the number of molecule annotated per document in both PatCID and Reaxys for our random benchmark (D2C-RND). It illustrates that (1) around 33% of documents have 4 times more annotations in PatCID than in Reaxys, (2) around 10% of documents has 4 times more annotations in Reaxys than in PatCID (these annotations probably come from the text), and (3) most documents have similar number of annotations in both databases. For each case, we also added a page from an example document. Additionally, we noticed that patents for which PatCID has an advantage are distributed across the different offices and publication year slices.

3. How to ensure the correctness of the molecule structure in the database constructed from the patents? Is there a filtering step, or the quality control of the database, or things were mainly based on the recognition precision of the pipeline?

Answer: The PatCID dataset is created fully automatically and without human intervention, as the automatically-created databases Google Patents and SureChEMBL. However, our evaluation of molecular and document search capabilities on the random benchmark (D2C-RND) and uniform benchmark (D2C-UNI) can be considered as a step of quality control. It allows us to assess the correctness of structures stored in PatCID.

In the recognition pipeline itself, there are some simple filtering steps such as verifying that the generated structures contain only one fragment. These points were added in the “Results - Document Ingestion Pipeline” section.

4. Can the proposed method be used for scientific papers (e.g. in chemistry or bioinformatics) too? The papers containing structure images may also provide valuable data.

Answer: We thank reviewer #3 for this suggestion. The proposed pipeline can indeed be used to process scientific papers in chemistry or bioinformatics. We provide an example in the GitHub repository [1] (see “Additional Visualization” section in the README).

References

- [1] Morin, L., Weber, V., Meijer, G. I., Yu, F. & Staar, P. W. J. PatCID GitHub. <https://github.com/DS4SD/PatCID> (2024).
- [2] Papadatos, G. et al. SureChEMBL: a large-scale, chemically annotated patent document database. *Nucleic Acids Research* 44, D1220–D1228 (2015).
- [3] Lawson, A. J., Swienty-Busch, J., Géoui, T. & Evans, D. The Making of Reaxys—Towards Unobstructed Access to Relevant Chemistry Information, chap. 8, 127–148 (American Chemical Society, 2014).
- [4] Gabrielson, S. W. SciFinder. *J. Med. Libr. Assoc.* 106 (2018).
- [5] Zhou, C., Liu, W., Song, X., Yang, M. & Peng, X. YoDe-Segmentation: automated noise-free retrieval of molecular structures from scientific publications. *Journal of Cheminformatics* 15, 111 (2023).
- [6] Tang, B. al. Automated molecular structure segmentation from documents using ChemSAM. *J Cheminform* 16, 29 (2024).
- [7] Jurriaans, T. et al. One Strike, You're Out: Detecting Markush Structures in Low Signal-to-Noise Ratio Images. *ArXiv* (2023). <https://arxiv.org/abs/2311.14633>.
- [8] Morin, L. et al. MolGrapher: Graph-based Visual Recognition of Chemical Structures. In *Proceedings of the IEEE/CVF International Conference on Computer Vision (ICCV)*, 19552–19561 (2023).
- [9] Senger, S., Bartek, L., Papadatos, G. & Gaulton, A. Managing expectations: assessment of chemistry databases generated by automated extraction of chemical structures from patents. *Journal of Cheminformatics* 7, 49 (2015).
- [10] Google Patents, US9096558. <https://patents.google.com/patent/US9096558B2/en?q=US9096558> (Accessed: May 2024).
- [11] Google Patents, US20220127225. <https://patents.google.com/patent/US20220127225A1/en?q=US20220127225> (Accessed: May 2024).
- [12] Ohms, J. Current methodologies for chemical compound searching in patents: A case study. *World Patent Information* 66, 102055 (2021).
- [13] “Reaxys is a web-based tool for the retrieval of information about chemical compounds and data from published literature”. <https://en.wikipedia.org/wiki/Reaxys> (Accessed: May 2024).
- [14] “SciFinder is a database of chemical and bibliographic information”. https://en.wikipedia.org/wiki/Chemical_Abstracts_Service (Accessed: May 2024).
- [15] Wang, J. et al. Multi-modal chemical information reconstruction from images and texts for exploring the near-drug space. *Briefings in Bioinformatics* 23, bbac461 (2022).

REVIEWER COMMENTS

Reviewer #1 (Remarks to the Author):

The authors fully implemented my suggestions for changes to the manuscript. I have no further objections.

Reviewer #1 (Remarks on code availability):

None

Reviewer #2 (Remarks to the Author):

The authors have clearly considered the reviewer comments carefully and have made significant efforts to address the points raised. Overall the changes to the accessibility of the material and clarity of the manuscript are a great improvement. The manuscript reads more clearly and the end result will be more useful to a broader audience. Thank you to the authors for their efforts.

I have only one minor remark about their changes: it appears, based on the new information in the methods section, that the overlap between PatCID and PubChem was performed using canonical SMILES - however if these "canonical SMILES" are not generated with exactly the same algorithm, these are not comparable and the overlap could be an underestimate (indeed, if SMILES were used I'm somewhat surprised the numbers are as high as they were - yet the methods section does not mention the software PubChem use to generate their SMILES). Does this number hold true if InChIKey first block is used? The InChIKey file is also available for bulk download on PubChem. InChIKey is consistent across toolkits and this more robust - and this would be somewhat more consistent with their other calculations. Otherwise, if the exact same SMILES canonicalization algorithm, or another form of standardization was used prior to the comparison, this should be stated so this is clear for readers.

Reviewer #3 (Remarks to the Author):

The revised manuscript has made responses to my previous comments. There are a few comments I would like to discuss further, as below.

(1) The authors, when evaluating the document segmentation, image classification, and molecule recognition performance, introduced two benchmark datasets: D2C-RND (Document to Chemical Structures, Random) and D2C-UNI (Document to Chemical Structures, Uniform). I am wondering why you need to use new data for evaluation since each of the three tasks should be pretty common and with benchmark datasets already available. Clarifications are needed on this point.

(2) "PatCID contains 81M chemical-structure images and 14M unique chemical structures." Please in the abstract make clear what is meant by chemical structures (e.g., SMILES)

(3) D2C-RND and D2C-UNI only contain a few hundred chemical structures. Would this be enough large to be used to evaluate the quality of annotations in PatCID dataset? please elaborate.

(4) "PatCID can be used to search for patents containing specific molecules or, find which molecules appear in which patents". For the retrieval part, the authors only reported metrics like recall; however for PatCID to be practically useful as a database, it would also be useful to discuss the retrieval efficiency (such as the time taken to find the targets for each query input) in comparison with other methods (if any). As to "find which molecules appear in which patents", this seems trivial and can you add a bit more on its definition (to me it's like browse what are there in

a document)? Do you list all the molecules in a specific document?

(5) " PatCID even competes with proprietary manually-curated patent-databases." - such a statement in the abstract lacks detailed explanations. what are the manual datasets? are they widely used? how large are they? the advantages are regarding which metric, etc.

(6) It would also be useful to add information to abstract like the file-size of the dataset and the format.

June 19, 2024

Dear reviewers,

Thank you for your positive feedback and the suggestions for improvements of the manuscript "PatCID: an open-access dataset of chemical structures in patent documents" (NCOMMS-24-09985B).

Reviewers' questions are addressed in detail below. The corresponding modifications in the manuscript are marked **in blue** (in supplementary files).

Reviewer #1

The authors fully implemented my suggestions for changes to the manuscript. I have no further objections.

Answer: Thank you.

Reviewer #2

The authors have clearly considered the reviewer comments carefully and have made significant efforts to address the points raised. Overall the changes to the accessibility of the material and clarity of the manuscript are a great improvement. The manuscript reads more clearly and the end result will be more useful to a broader audience. Thank you to the authors for their efforts.

I have only one minor remark about their changes: it appears, based on the new information in the methods section, that the overlap between PatCID and PubChem was performed using canonical SMILES - however if these "canonical SMILES" are not generated with exactly the same algorithm, these are not comparable and the overlap could be an underestimate (indeed, if SMILES were used I'm somewhat surprised the numbers are as high as they were - yet the methods section does not mention the software PubChem use to generate their SMILES). Does this number hold true if InChIKey first block is used? The InChIKey file is also available for bulk download on PubChem. InChIKey is consistent across toolkits and this more robust - and this would be somewhat more consistent with their other calculations. Otherwise, if the exact same SMILES canonicalization algorithm, or another form of standardization was used prior to the comparison, this should be stated so this is clear for readers.

Answer: Canonical SMILES of PatCID and PubChem are(!) generated using the same algorithm. Instead of directly using the PubChem-provided SMILES, we canonicalize these SMILES using the same algorithm as used for SMILES in PatCID (using RDKit [1]). Thus, the suggested comparison with InChIKey should not provide additional information. We clarified this point in the "Method - Databases Querying" section of the manuscript.

Reviewer #3

The revised manuscript has made responses to my previous comments. There are a few comments I would like to discuss further, as below.

(1) The authors, when evaluating the document segmentation, image classification, and molecule recognition performance, introduced two benchmark datasets: D2C-RND (Document to Chemical Structures, Random) and D2C-UNI (Document to Chemical Structures, Uniform). I am wondering why you need to use new data for evaluation since each of the three tasks should be pretty common and with benchmark datasets already available. Clarifications are needed on this point.

Answer: While the three individual tasks indeed seem common, there actually are no valid benchmarks to evaluate (1) the segmentation and (2) the classification. For a different purpose (non-patent literature), the DECIMER-Segmentation [2] model was evaluated on a benchmark, but without bounding-box annotations. Benchmarks are available to evaluate (3) the recognition; this evaluation is available in our separate publication [3] (in total more than 40 000 chemical structures are evaluated). In this manuscript, we therefore introduce the two benchmark datasets D2C-RND and D2C-UNI to evaluate simultaneously all three components of a document to chemical-structure pipeline. We clarified this in the “Results - Document Ingestion Pipeline” section of the manuscript.

(2) "PatCID contains 81M chemical-structure images and 14M unique chemical structures." Please in the abstract make clear what is meant by chemical structures (e.g., SMILES)

Answer: Nature Communications advises against using acronyms in the abstract. We therefore prefer to mention SMILES starting in the “Result - Data Statistics” section.

(3) D2C-RND and D2C-UNI only contain a few hundred chemical structures. Would this be enough large to be used to evaluate the quality of annotations in PatCID dataset? please elaborate.

Answer: The number of chemical structures that would need to be manually annotated to be formally representative for the PatCID dataset is admittedly not within our available resources. Trying to overcome this limitation, we designed two independent sampling strategies to create D2C-RND and D2C-UNI. D2C-RND tries to approach a statistical randomization and additionally D2C-UNI represents even less abundant documents. (Our annotation tool is made available to allow further extension of D2C-UNI and D2C-RND benchmarks). We elaborated on this point in the “Method - Database Evaluation - Benchmarks and Metrics” section of the manuscript.

(4) "PatCID can be used to search for patents containing specific molecules or, find which molecules appear in which patents". For the retrieval part, the authors only reported metrics like recall; however for PatCID to be practically useful as a database, it would also be useful to discuss the retrieval efficiency (such as the time taken to find the targets for each query input) in comparison with other methods (if any). As to "find which molecules appear in which patents", this seems trivial and can you add a bit more on its definition (to me it's like browse what are there in a document)? Do you list all the molecules in a specific document?

Answer: Based on our internal implementation of the database (using a vector database for molecular structures), a typical search takes order-of-magnitude one second. We don't want to make a claim on being better or worse than Reaxys and SciFinder databases. (A naive string search in the PatCID dataset would take order-of-magnitude 10 seconds). This information was added to the GitHub repository.

For a specific patent document, PatCID can indeed list all molecules in the document. This is particularly relevant when assessing the intellectual-property landscape of, for example, a competitor. We reformulated this sentence in the introduction.

(5) " PatCID even competes with proprietary manually-curated patent-databases." - such a statement in the abstract lacks detailed explanations. what are the manual datasets? are they widely used? how large are they? the advantages are regarding which metric, etc.

Answer: Due to word-count limitation of the abstract, we decided to include this arguably more detailed information in the introduction section (“what are the manual datasets?”, “are they widely used?”), and the result section (“which metric?”). Regarding “how large are they?”, the number of compounds in Reaxys and SciFinder is not publicly available, as we indicate in Table 1.

(6) It would also be useful to add information to abstract like the file-size of the dataset and the format.

Answer: Due to word-count limitation of the abstract, we decided to add this arguably more detailed information in the GitHub repository [4].

References

- [1] Landrum, G. et al. RDKit: Open-Source Cheminformatics Software. <http://www.rdkit.org/> 542 (2006).
- [2] Rajan, K., Brinkhaus, H. O., Sorokina, M., Zielesny, A. & Steinbeck, C. DECIMER-Segmentation: Automated extraction of chemical structure depictions from scientific literature. *Journal of Cheminformatics* 13, 20 (2021).
- [3] Morin, L. et al. MolGrapher: Graph-based Visual Recognition of Chemical Structures. In Proceedings of the IEEE/CVF International Conference on Computer Vision (ICCV), 19552–19561 (2023).
- [4] Morin, L., Weber, V., Meijer, G. I., Yu, F. & Staar, P. W. J. PatCID GitHub. <https://github.com/DS4SD/PatCID> (2024).

REVIEWERS' COMMENTS

Reviewer #2 (Remarks to the Author):

Thank you to the authors for considering the reviewer comments (again!), my additional question arising from the last review was clarified appropriately by the updates the authors made to the manuscript (and the response) and it seems to me that their responses and corresponding updates in light of the other reviewer comments were also reasonable.

Reviewer #3 (Remarks to the Author):

The authors have addressed my previous concerns.